# A Learning-Augmented Dynamic Programming Approach for Orienteering Problem with Time Windows

**Guansheng Peng**[1,2]    **Lining Xing**[3]    **Fuyan Ma**[4]    **Guopeng Song**[5]*
**Aldy Gunawan**[6]    **Pieter Vansteenwegen**[7]
[1]School of Advanced Engineering, Great Bay University, China
[2]Shenzhen International Center For Industrial And Applied Mathematics,
Shenzhen Research Institute of Big Data, China
[3]Research Institute of Intelligent Control and Manufacturing System,
Jiangsu University of Technology, China
[4]Defense Innovation Institute, Chinese Academy of Military Science, China
[5]College of Systems Engineering, National University of Defense Technology, China
[6]School of Computing and Information Systems, Singapore Management University, Singapore
[7]KU Leuven Institute for Mobility – CIB, KU Leuven, Belgium
pengguansheng@gbu.edu.cn, guopeng.song@nudt.edu.cn

## Abstract

Recent years have witnessed a surge of interest in solving combinatorial optimization problems (COPs) using machine learning techniques. Motivated by this trend, we propose a learning-augmented exact approach for tackling an NP-hard COP, the Orienteering Problem with Time Windows, which aims to maximize the total score collected by visiting a subset of vertices in a graph within their time windows. Traditional exact algorithms rely heavily on domain expertise and meticulous design, making it hard to achieve further improvements. By leveraging deep learning models to learn effective relaxations of problem restrictions from data, our approach enables significant performance gains in an exact dynamic programming algorithm. We propose a novel graph convolutional network that predicts the directed edges defining the relaxation. The network is trained in a supervised manner, using optimal solutions as high-quality labels. Experimental results demonstrate that the proposed learning-augmented algorithm outperforms the state-of-the-art exact algorithm, achieving a 38% speedup on Solomon's benchmark and more than a sevenfold improvement on the more challenging Cordeau's benchmark.

## 1 Introduction

Combinatorial optimization problems (COPs) arise in a wide range of application domains, such as logistics, telecommunications, manufacturing, and finance. Solving these problems efficiently can result in significant cost reductions. Nevertheless, their NP-hard nature presents major computational challenges. To handle these challenges, traditional approaches, including exact and heuristic methods, have been developed over the past decades.

Recently, machine learning (ML) has emerged as a complementary approach, addressing the limitations of traditional approaches through data-driven learning [4]. A variety of studies adopt an end-to-end framework that directly outputs solutions. Among these, graph neural network models has been successfully applied to COPs defined over graphs, such as the Traveling Salesperson Problem

---

*Correspondence

39th Conference on Neural Information Processing Systems (NeurIPS 2025).

(TSP), Minimum Vertex Cover, Maximum Cut, and others [9, 24, 15], as they naturally operate on the graph structures inherent to these problems. Additionally, sequence-to-sequence learning methods, such as Pointer Networks [3], Graph Attention Networks [19] and Transformers [6], offer an autoregressive approach for solving COPs. Although the above studies have achieved some success, there are still limitations. First, these learning models are rarely applied to complex and higher-dimensional COPs which usually requires hard-won explicit expertise for higher performance. In many cases, the end-to-end learning approaches underperform traditional approaches [1]. Second, most existing studies focus on combining ML with heuristics, while integrations with exact algorithms remain largely unexplored. The predominant trend of this direction is to formulate COPs as mixed-integer programmings (MIPs) and enhance key components of branch-and-bound (B&B) based algorithms for MIPs using ML techniques [31]. ML techniques are leveraged to guide primal heuristics [11, 17], make high-quality branching decisions [13, 27], separate cutting planes to strengthen the formulation [10], and improve node selection strategies [14, 22] within the B&B framework. The advantage of this direction is that the ML techniques accelerate the traditional exact algorithms without compromising the exactness.

Our work focuses on a deep integration of ML with dynamic programming (DP), another exact solving paradigm for COPs. Unlike MIP-based approaches, DP solves COPs by explicitly defining a state-space formulation and determining state transition recurrences based on the problem's specific structure. DP is widely used to efficiently solve COPs with clear problem structures, such as knapsack and routing problems. However, its main drawback is the risk of state space explosion, also known as the 'curse of dimensionality'. To address this limitation, we employ ML techniques to approximate a projection function that maps the original state space onto a lower-dimension space, thereby substantially reducing its dimensionality. To the best of our knowledge, such a learning-based paradigm for COPs has not been explored in the existing literature. Numerical experiments demonstrate its superior performance compared with state-of-the-art (SOTA) exact algorithms.

To illustrate this paradigm, this paper focuses on a classic COP, the Orienteering Problem with Time Windows (OPTW). In the OPTW, a set of vertices is given, each associated with a score, a service time and a time window. The objective is to determine a tour that maximizes the total collected score while respecting all time window constraints. This problem, also known as the Selective TSP [23], has more complex resource constraints and decision variables than the classic TSP. To solve the problem efficiently, we propose a learning-augmented exact approach, DP-NG-ML, which combines an advanced deep learning model with a DP method enhanced by a so-called Ng-route Relaxation (DP-NG). This learning model, termed DiConvNet, accelerates the DP algorithm by approximating effective relaxations of the problem's elementarity restrictions. The elementarity restrictions ensure that each vertex is visited at most once in a tour. Relaxing the restrictions reduces the dimensionality of the searched state space but may lead to a non-elementary path. The main contributions of this paper are summarized as follows:

- **Problem Perspective:** Compared to the well-studied TSP, the problem we address is a more intricate COP, involving more constraints and high-dimensional features. Whether advanced learning models can achieve comparable success on such complex problems remains an open question. Moreover, we claim that OPTW applications can particularly benefit from the use of ML techniques. For example, the Tourist Trip Design Problem (TTDP) [33], a variant of the OPTW, plans personalized itinerary for each individual tourist in a city within time limits. In practice, planners in the same city often encounter similar instances, with variations mainly in tourist preferences. Such consistency provides an opportunity for ML to uncover regional patterns and enhance effective real-time decision-making.

- **Methodological Innovation:** Our method combines the ML techniques and transitional DP algorithm, preserving optimality while significantly enhancing performance through data-driven insights. By leveraging imitation learning, the model accurately approximates the state space relaxation, thereby accelerating the DP algorithm. To the best of our knowledge, this is the first work to propose such a solution paradigm. Despite the state space relaxation method is problem-specific, the overall paradigm is anticipated to generalize across a broad spectrum of COPs defined over graph structures.

- **Empirical Validation:** Extensive experiments on benchmark instances show that the proposed learning-augmented exact algorithm, DP-NG-ML, significantly outperforms the SOTA exact algorithm, the pulse algorithm [12]. Moreover, this algorithm demonstrates a considerable speedup compared to the version without the integrated learning model. The experiments also show that the learning model achieves high accuracy in predictions, outperforming the heuristic methods commonly used in the literature on COPs.

## 2 Related Work

The OPTW is an NP-hard problem and several heuristics have been proposed to obtain high-quality solutions, including a depth-first search tree algorithm [16], a granular variable neighborhood search [21], an ant colony system algorithm [26] and an iterated local search algorithm [34]. Ricardo Gama and Hugo L. Fernandes [29] proposed the first work to tackle the OPTW using a ML approach, based on a reinforcement learning framework combined with Pointer Networks. These methods are empirically efficient but offer no guarantee of exactness or optimality gap.

Due to the complexity of the OPTW, only two exact approaches have been proposed: a bidirectional DP with Decremental State Space Relaxation (DSSR) [30] and a SOTA pulse algorithm [12]. The former [30] solves a relaxed version of the OPTW via a DP algorithm, allowing a subset of vertices to be visited more than once. The latter [12], which is based on a branch and bound search scheme with efficient pruning strategies, significantly outperforms the former on the OPTW benchmark in terms of both solution quality and computational time. We adopt the same DP iterative framework as in [30] but replace the DSSR with a more advanced relaxation technique, ng-route relaxation [2]. This relaxation is guided by so-called ng-sets, which impose the elementarity restrictions over a set of edges and are approximated by our proposed learning model. To the best of our knowledge, no existing work has attempted to estimate the state space relaxation in DP using ML.

## 3 Dynamic programming with state space relaxation for OPTW

### 3.1 Problem definition and basic DP formulation

The OPTW is defined on a directed graph $G = (V, E)$, where $V$ is the set of vertices and $E$ is the set of directed edges representing accessibility between vertex pairs. The cardinalities of $V$ and $E$ are denoted as $|V|$ and $|E|$, respectively. For each vertex $i \in V$, we are given: a 2D coordinate $(x_i^1, x_i^2)$, a positive score $s_i$ collected upon visiting the vertex, a time window $[w_i^1, w_i^2]$ specifying the feasible arrival time interval, and a non-negative service duration $d_i$ representing the time required to serve the vertex. For each pair of vertices $(i, j)$, the travel time $t_{ij}$ is equal to the Euclidean distance between them. The objective is to determine an optimal tour that starts and ends at the depot 0, visits a subset of vertices in $V$, maximizes the collected scores, and satisfies all time window constraints. This tour must be elementary, meaning it contains no cycles or duplicate vertices. The maximum length of the route is limited to a given time budget $T_{max}$.

The OPTW can be formulated as a special case of the classic Resource constrained Elementary Shortest Path Problem [30], for which no polynomial-time algorithm is known. DP is considered an efficient exact approach for solving this problem. Let $F(S, \tau, i)$ denote the maximum collected score of a path starts from the depot and ends at vertex $i$, visiting each vertex in set $S$ exactly once, with an elapsed time of $\tau$. $F(S, \tau, i)$ can be computed by solving the recurrence equation:

$$F(S, \tau, i) = \max_{(j,i) \in E} \{F(S \setminus \{i\}, \tau', j) + s_j | \tau + t_{ji} \leq w_i^2, w_j^1 \leq \tau' \leq w_j^2\},$$

$$\forall i \in V, S \subseteq V, w_i^1 \leq \tau \leq w_i^2. \tag{1}$$

Set $S$ denotes the set of visited vertices and can be interpreted as a set of dummy resources, each with a unit capacity. This set is typically encoded as a binary vector of length $|V|$. By recording this set, the path can be extended to other unvisited vertices so that cycles are avoided and all the searched paths are elementary. It is evident that the size of the state-space graph grows exponentially with the number of vertices $|V|$. To reduce the number of states explored, we adopt a state space relaxation technique, namely the ng-route relaxation Baldacci et al. [2] to project the original state space $(S, \tau, i)$ onto a lower-dimensional space.

### 3.2 Ng-route relaxation

The ng-route relaxation, introduced by Baldacci et al. [2], provides a good compromise between enforcing elementarity constraints and enabling efficient exploration of the state space. For each vertex $i \in V$, we define an ng-set $N_i \subseteq V$, which is a selected subset of vertices associated with vertex $i$ (according to some criterion). Let $P = \{0, i_1, ..., i_p\}$ be a partial path starting from the depot to vertex $i_p$, which is associated with the set of visited vertices $\mathcal{S}(P)$ and an elapsed time $\tau$. The

basic DP algorithm explores the state space by extending paths to all possible succeeding vertices using equation (1). We define $\Pi(P) \subseteq \mathcal{S}(P)$ as the "memory" of path $P$. During the past extension of path $P$, for any intermediate vertex $i_k (1 \le k \le p)$, only the visited vertices that belong to the ng-set of $i_k$ can be retained in memory of $P$. Therefore, the memory set $\Pi(P)$ is defined as the intersection of the visited vertices and the ng-sets of all subsequently visited vertices, and is given by:

$$\Pi(P) = \{i_k \in \mathcal{S}(P) \setminus \{i_p\} : i_k \in \cap_{q=k+1}^p N_{i_q}\} \cup \{i_p\}. \tag{2}$$

With this definition, path $P$ cannot be extended to vertices which are included in its memory $\Pi(P)$. The predefined ng-sets determine the degree of state space relaxation. If all ng-sets are empty, all elementarity constraints are relaxed; conversely, if $N_i = V$ for every vertex $i \in V$, the algorithm is equivalent to the original one. If vertex $i_k$ does not belong to $N_{i_p}$, it is not recorded in the memory of path $P$ and may be revisited by the next extension of $P$. To prevent such cycles, $i_k$ is added to $N_{i_p}$, thereby tightening the relaxation.

By applying this relaxation, the original state $(\mathcal{S}(P), \tau, i)$ of path $P$ is mapped onto a lower-dimensional state $(\Pi(P), \tau, i)$, where $\Pi(P)$ is represented as a binary vector of length $|N_i|$. Since $|N_i| \le |V|$ for each vertex $i$, the size of the state-space graph is significantly reduced. Figure 1 illustrates the path extension using given ng-sets. The main drawback of the relaxation technique is that the search in the relaxed state space does not guarantee to find optimal solution. A practical compromise is to iteratively tighten the relaxation, i.e., enlarge the ng-sets, based on the optimal non-elementary path obtained in each iteration of DP, until an optimal elementary path is eventually found. This procedure will be detailed in the next subsection.

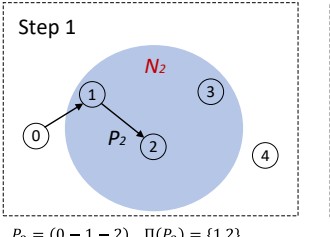 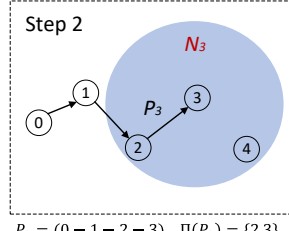 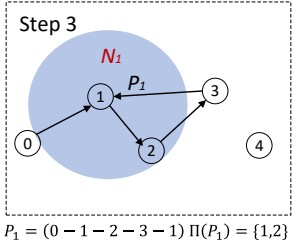

Figure 1: Illustration of path extensions using ng-sets. The blue circular area indicates the ng-set of the associated vertex. Path $P_2$ is extended to vertex 3 to produce new path $P_3$, as $3 \notin \Pi(P_2)$. The memory of path $P_3$ forgets the visited vertex 1, as $N_3$, does not contain 1. Consequently, vertex 1 can be revisited by path $P_3$.

Note that this iterative framework may incur higher computational costs compared to the basic DP algorithm, as the dimensionality of the state space increases with each iteration. An inappropriate criterion for expanding the ng-sets may introduce "unnecessary" vertices, further inflating the search space. We demonstrate that this issue can be mitigated by leveraging ML techniques to predict the ng-sets based on hidden patterns extracted from data. Moreover, the data-driven approach serves as an effective heuristic to initialize the ng-sets, thereby reducing the number of required iterations. Essentially, our goal is to train a neural network that approximates the projection function, mapping the original state space to a suitable lower-dimension space in a one-shot manner. In this case, the ng-set of vertex $i$ can be viewed as a set of directed edges originating from $i$.

### 3.3 DP with ng-route relaxation

A state of path $P$ can be interpreted as a label $L(P) = (i, \tau(P), s(P), \Pi(P))$, where $i$ is the last visited vertex and $s(P)$ is the collected score. Extending label $L(P)$ to vertex $j$ indicates adding $j$ to the end of $P$, thereby generating a new label. This extension is feasible only if the time window constraint at $j$ is satisfied and $j$ has not been visited by path $P$, i.e., $j \notin \Pi(P)$. To limit the exponential growth in the number of labels, a dominance test is applied to identify and discard labels that cannot lead to an optimal solution. By exploring the state space defined by the current ng-sets, the path with the highest scores is considered the optimal path for that iteration. If the obtained path is non-elementary, the vertices involved in a cycle is updated to include the duplicate vertex, thereby preventing the same cycle from occurring in the next iteration. This exact approach is referred to as the DP-NG algorithm, with a detailed description given in Appendix A.1 and an outline in Appendix

A.2. Figure 2 illustrates an example of the DP-NG algorithm applied to a seven-vertex instance. It can be expected that states with lower time consumption and higher collected scores are more likely to survive the dominance test. Consequently, pairs of vertices with high individual scores, close spatial proximity and highly overleaped time windows are more likely to form cycles. This observation motivates the use of graph structural features for predicting the ng-sets.

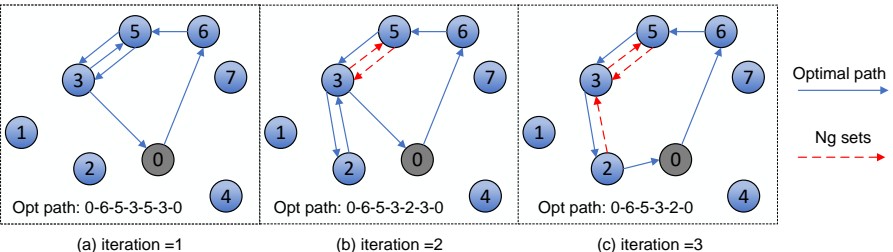

(a) iteration =1          (b) iteration =2          (c) iteration =3

Figure 2: Illustration of the DP-NG algorithm at different iterations. The blue directed arrows represent the edges of the optimal path in the current iteration, while the red dotted arrows indicate the current ng-sets. The target vertex of the red dotted arrow belongs to the ng-set of its source vertex. In the first iteration, the DP algorithm finds a non-elementary optimal path (0-6-5-3-5-3-0), and subsequently adds the edges (5,3) and (3,5) to the ng-sets to avoid revisiting the cycle between vertices 5 and 3 in future iterations. In the second iteration, it returns another non-elementary optimal path (0-6-5-3-2-3-0), and edge (2,3) is added to the ng-sets. The algorithm ultimately finds an elementary path (0-6-5-3-2-0), at which point it terminates.

Despite its theoretical merits, the ng-route relaxation may result in longer computation times. In the worst case, as iterations increase, the ng-sets may eventually include all vertices, effectively reducing the method to standard DP without relaxation. However, if the ng-sets can be accurately predicted, the number of required iterations can be significantly reduced. This insight drives the development of our learning approach.

# 4 Learning approach for state space relaxation

## 4.1 Framework

The overall framework of our proposed learning-augmented approach is illustrated in Figure 3. To enhance prediction accuracy, we first apply an edge reduction procedure that tags edges unlikely to be part of any ng-set (see Appendix B.1 for details). The vertex and edges features are fed into to the DiConvNet model which ultimately outputs a probabilistic heatmap over the edges. This heatmap indicates the likelihood of each edge being included in the ng-sets. A subset of edges with the highest probabilities is then selected to initialize the ng-sets, and the DP-NG algorithm is executed to compute the optimal path.

Predicting the ng-sets for a graph can be formulated as a binary classification problem over all possible directed edges, with those in the ng-sets labeled as positive. We train the DiConvNet network $f$ in a supervised manner, using labels generated by the proposed DP-NG algorithm.

## 4.2 Edge reduction

To improve prediction accuracy, we propose an edge reduction procedure that leverages problem-specific characteristics to exclude edges that cannot belong to ng-sets. Two types of edges are excluded: (1) the edges connected to the depot or self-loops; (2) any edge $e_{ij}$ satisfying the following condition:

$$max\{w_j^1 + d_j + t_{ji}, w_i^1\} + d_i + t_{ij} > w_j^2. \tag{3}$$

The first type of edges is excluded by definition, while the second type corresponds to a situation where cycle $(j - i - j)$ is infeasible due to the time window constraints.

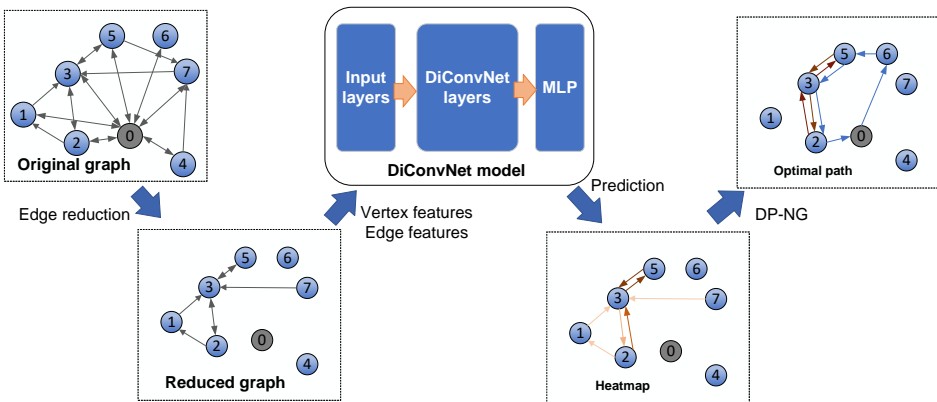

Figure 3: Framework of the DP-NG-ML algorithm.

## 4.3 DiConvNet model

The DiConvNet model is based on the Graph Convolutional Network (GCN) architecture, ConvNet, originally proposed by Bresson and Laurent [5]. ConvNet was designed to directly approximate solutions to the TSP and has demonstrated strong performance [15]. However, preliminary experiments indicate that this model performs poorly for our problem, primarily because it is tailored for undirected graphs. To accommodate the asymmetry inherent in directed graphs and problems with time windows, we develop a directed variant of this model, referred to as DiConvNet.

The DiConvNet model comprises three main components: input layers, DiConvNet layers, and a Multi-Layer Perceptron (MLP) classifier. The input layers project the initial vertex and edge features into $h$-dimensional embeddings, which are refined by DiConvNet layers. The MLP classifier then estimates the probability of each edge being included in the ng-sets. Details of the MLP classifier and the loss function are provided in Appendix B.1.

**Input Layer** The input features for each vertex $i$ include its 2D coordinates $x_i = (x_i^1, x_i^2)$, score $s_i$, service time $d_i$, and time window $w_i = (w_i^1, w_i^2)$. These features are then embedded to $h$-dimensional vectors:

$$\alpha_i = \theta_1(x_i||s_i||d_i||w_i) + b_1, \tag{4}$$

where $\theta_1 \in \mathbb{R}^{h \times 6}$ and $\cdot||\cdot$ is the concatenation operator. We consider the Euclidean distance $t_{ij}$ and the edge tags $c_{ij} \in \{0, 1, 2, 3\}$ as the input features for each edge $e_{ij} \in E$. Tags 1 and 0 indicate whether an edge is included or not in the corresponding reduced graph, while tags 2 and 3 represent edges connected to the depot and self-loops, respectively. Each of the two edge features is independently embedded into a $\frac{h}{2}$-dimensional vector, and the resulting vectors are concatenated to form the final edge embedding, as expressed below:

$$\beta_{ij} = \theta_2 t_{ij} + b_2 || \theta_3 c_{ij}, \tag{5}$$

where $\theta_2 \in \mathbb{R}^{\frac{h}{2} \times 1}$, $\theta_3 \in \mathbb{R}^{\frac{h}{2} \times 4}$.

**DiConvNet layer** We define a source embedding $p_i$ and a target embedding $q_i$ for each vertex $i$, and an edge embedding $h_{ij}$ for edge $e_{ij}$. The source embedding $p_i$ is updated by iteratively aggregating the target embeddings of its out-neighbors (denoted as $\mathcal{N}_{out}(i)$), while the target embedding $q_i$ is updated by aggregating the source embeddings of its in-neighbors (denoted as $\mathcal{N}_{in}(i)$). This directional propagation mechanism enables the model to effectively capture the asymmetric structural patterns in directed graphs. Consequently, the edge embeddings, derived from the source and target embeddings, are asymmetric. Both the source and target embeddings of each vertex $i$ are initialized with $\alpha_i$, while the initial embedding of each edge $e_{ij}$ is set to $\beta_{ij}$. The embeddings are updated at each iteration as:

$$p_i^{l+1} = p_i^l + \text{ReLU}(\text{BN}(\theta_4^l p_i^l + \sum_{j \in \mathcal{N}_{out}(i)} \eta_{ij}^l \odot \theta_5^l q_j^l)) \text{ with } \eta_{ij}^l = \frac{\sigma(h_{ij}^l)}{\sum_{j' \in \mathcal{N}_{out}(i)} \sigma(h_{ij'}^l) + \epsilon}, \tag{6}$$

$$q_i^{l+1} = q_i^l + \text{ReLU}(\text{BN}(\theta_6^l p_i^l + \sum_{j \in \mathcal{N}_{in}(i)} \eta_{ji}^l \odot \theta_7^l p_j^l)) \text{ with } \eta_{ji}^l = \frac{\sigma(h_{ji}^l)}{\sum_{j' \in \mathcal{N}_{in}(i)} \sigma(h_{j'i}^l) + \epsilon}, \quad (7)$$

$$h_{ij}^{l+1} = h_{ij}^l + \text{ReLU}(\text{BN}((\theta_8^l h_{ij}^l + \theta_9^l p_i^l + \theta_{10}^l q_i^l)), \quad (8)$$

where $\theta_4, \theta_5, \theta_6, \theta_7, \theta_8 \in \mathbb{R}^{h \times h}$, $\sigma$ is the sigmoid function, $\epsilon$ is a small constant, $\odot$ represents the Hadamard product, ReLU is the rectified linear unit, and BN stands for batch normalization. $\eta_{ij}^l$ acts as edge gate to govern the propagation of information from edge embeddings to vertex embeddings during each layer. Unlike the approach of Joshi et al. [15], our model aggregates messages from all incoming edges to each vertex. This broader aggregation scheme enables the network to capture local structural information more comprehensively. Figure 4 illustrates how the DiConvNet layer updates the vertex embeddings and asymmetric edge embeddings. During each iteration, the source, target and edge embeddings are updated in sequence.

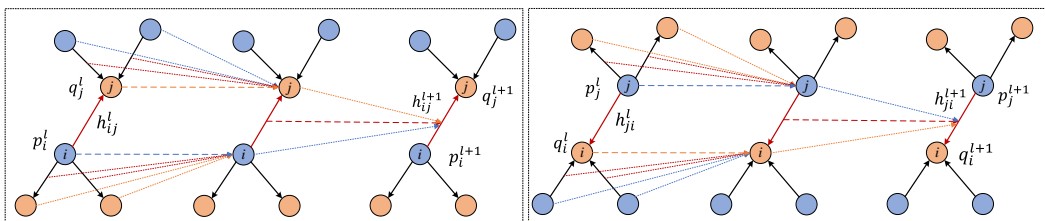

Figure 4: The proposed DiConvNet layer for updating source, target and edge embeddings for each vertex $i, j$ and edge $e_{ij}$ in the graph. Blue and orange circles represent source and target embeddings, respectively, with dotted arrows indicating their propagated messages. Red dotted arrows represent messages propagated by edge embeddings.

## 5 Experiments

### 5.1 Experimental setup

**Datasets.** There are two main classes of OPTW benchmark instances in the literature: Solomon's instances[32] and Cordeau's instances [7, 8], both of which have known optimal solutions. In Solomon's instances, distances between locations are rounded to one decimal place, whereas in Cordeau's instances, they are rounded to two decimal places. We randomly generate separate sets of training and validation instances for each class, following the methodologies outlined in the corresponding literature. Finally, the benchmark instances are used to evaluate the proposed DP-NG and DP-NG-ML algorithms. Detailed parameter settings are provided in Appendix C.

- **Solomon's instances.** This dataset contains 29 instances with 100 vertices, organized into three categories: r-instances, where vertices are randomly located; c-instances, where vertices are clustered; and the rc-instances, which combine random and clustered vertices. Within each category, the instances share identical vertex coordinates, service times, and scores, differing only in the time window settings. Hence, we generate 10,000 training instances and 1,000 validation instances per category, using the same vertex information except for the time windows. However, since Solomon [32] does not provide the parameters of generating time windows, we empirically determine them and generalize the trained model to Solomon's benchmark instances.

- **Cordeau's instances.** This testbed comprises 10 instances, with the number of vertices ranging from 48 to 288. Compared to Solomon's instances, these instances are considered more challenging due to their wider time windows [30]. Moreover, Cordeau's benchmark instances vary substantially in size, making it difficult to generalize across different instance scales. We generate 100,000 training instances and 1,000 validation instances, each containing 50 vertices, and randomly select only 10,000 instances for each training epoch.

**Training.** In each training epoch, the 10,000 training instances are partitioned into 312 mini-batches, each comprising 32 instances. The model is trained using the Adam optimizer [18] with single-graph mini-batches, a learning rate of $10^{-4}$ and a decay factor of 1.01. The maximum training epochs is set to 1,500.

**Evaluation.** Since the performance of the DP-NG-ML algorithm is highly sensitive to both false positives and false negatives, we evaluate the learning model using the following metrics:

- **Error Rate:** the overall percentage of misclassified samples.
- **FNR:** the proportion of positive samples incorrectly classified as negative.
- **FPR:** the proportion of negative samples incorrectly classified as positive.

To evaluate performance on OPTW benchmark instances, we compare the proposed DP-NG and DP-NG-ML algorithms with the SOTA exact algorithm, the pulse algorithm [12]. Leveraging the publicly available implementation of the pulse algorithm[2] provided by Duque et al. [12], all algorithms are executed on the same computational platform to ensure a fair comparison. To assess the effectiveness of the proposed DiConvNet model, we compare it against the original ConvNet configured with identical architecture and training settings. The details of the ConvNet model are provided in Appendix B.2.

**Network settings.** For Solomon's instances, the DiConvNet model is configured with 10 DiConvNet layers, each containing 64 hidden units. For Cordeau's instances, the model uses 15 layers with 256 hidden units per layer. These configurations are chosen to enhance model performance while preventing memory overflow during training. The MLP classifier is configured with three layers across all cases.

## 5.2 Results

The learning models are trained on an NVIDIA Tesla V100-SXM2 GPU with 32 GB of memory. The DP-NG and DP-NG-ML algorithms, along with the pulse algorithm [12] for comparison, are executed on a desktop equipped with a 2.5 GHz Intel Core i5-13490F processor and 16 GB of RAM. All computational times are reported in seconds, denoted as "time(s)".

**Performance of predictive models.** Table 1 compares the performance of different predictive methods on Solomon's and Cordeau's benchmark instances. The best results are highlighted in bold. "Nearest" represents a widely adopted heuristic that constructs the ng-set of each vertex by selecting its $\tilde{n}$ closest neighbors [2], while the "MaxScores" heuristic, proposed in [28], incorporates spatial proximity, vertex scores and time window constraints in a more comprehensive manner. This comparison highlights that the formation of ng-sets involves intricate dependencies and structural patterns within the graph features. Such knowledge can be precisely learned through data-driven approaches.

Furthermore, the proposed DiConvNet achieves lower Error Rate and FPR than the original ConvNet model, although the ConvNet yields a lower FNR on the Solomon benchmark instances. Note that a slight increase in the FPR can lead to a substantial expansion in the size of the ng-sets, because of the large proportion of negative samples (edges) in the graph. The reduction in FPR can be attributed to the model's ability to capture asymmetric edge representations. For Cordeau's benchmark, although trained on random instances with only 50 vertices, the model still exhibits strong predictive capability. Nevertheless, to achieve further improvements, we only select the top $\tilde{n} \cdot |V|$ edges with the highest predicted probabilities, provided the probability exceeds $0.5$. These selected edges serve as the initial ng-sets, leading to improved performance of the DP-NG algorithm, as shown in Tables 2 and 3. The best configuration of DiConvNet for Cordeau's benchmark is shown in the last row of Table1, with detailed parameter tuning results presented in Appendix D.

The evaluation of the predictive models relies on labels derived from the current DP-NG algorithm. While further improvements to the label quality are possible through more sophisticated ng-set update strategies in the DP-NG algorithm, such enhancements may come with increased computational cost. For practical efficiency, we maintain the current implementation.

**Performance on OPTW benchmark.** Tables 2 and 3 report the performance of our proposed algorithms compared to the SOTA pulse algorithm [12] using Solomon's and Cordeau's benchmark instances. For Cordeau's benchmark, we set $\tilde{n} = 3$, while for Solomon's benchmark, we do not impose any limit on the size of the predicted ng-sets. All instances are solved to optimality by the exact algorithms, with the column "OptVal" reporting the corresponding optimal values. The sizes of the predicted and final ng-sets are listed in the columns "pre-ng-size" and "ng-size", respectively.

---

[2]`https://github.com/copa-uniandes/OPTW_Pulse.git`

Table 1: Comparison of different predictive methods on on Solomon's and Cordeau's benchmark instances.

| Predictive Method | Solomon's Benchmark | | | Cordeau's Benchmark | | |
|---|---|---|---|---|---|---|
| | Error Rate(%) | FNR(%) | FPR(%) | Error Rate(%) | FNR(%) | FPR(%) |
| Nearest ($\tilde{n}=2$) | 25.75 | 73.73 | 23.98 | 6.88 | 70.36 | 4.92 |
| MaxScores ($\tilde{n}=2$) | 24.44 | 51.42 | 23.29 | 5.89 | 53.99 | 4.4 |
| Nearest ($\tilde{n}=3$) | 29.17 | 66.66 | 27.72 | 9.10 | 61.50 | 7.59 |
| MaxScores ($\tilde{n}=3$) | 27.56 | 41.11 | 26.87 | 7.9 | 41.72 | 6.96 |
| Nearest ($\tilde{n}=4$) | 31.72 | 56.81 | 30.64 | 11.25 | 52.29 | 10.22 |
| MaxScores ($\tilde{n}=4$) | 30.24 | 35.42 | 29.86 | 10.11 | 32.67 | 9.62 |
| ConvNet | 5.46 | **9.97** | 5.33 | 10.60 | 2.74 | 10.72 |
| DiConvNet | 3.41 | 11.21 | 3.27 | 8.21 | **2.64** | 8.25 |
| DiConvNet($\tilde{n}=3$) | **3.25** | 12.09 | **3.06** | **4.49** | 6.40 | **4.40** |

The Column "iter." denotes the number of iterations required to compute the elementary optimal solution. The Column "$R_{pulse}$" reports the speedups of the DP-NG-ML algorithm compared to the pulse algorithm, while the column "$R_{ml}$" indicates the speedup of DP-NG-ML over the original DP-NG algorithm. Speedups greater than one are highlighted in bold. The last row presents the average results, where the average speedups are computed as the ratio of the computational time of the pulse algorithm to that of the DP-NG-ML algorithm.

Table 2 shows that, in 21 out of 29 instances, the learning-augmented DP-NG algorithm outperforms the pulse algorithm, despite achieving only a modest overall speedup of 1.26. The DP-NG algorithm becomes twice as fast when integrated with the learning model. Additionally, the number of iterations required to compute the elementary optimal solution is reduced from an average of 24.4 to 2.9.

Table 3 shows that the DP-NG-ML algorithm consistently outperforms the pulse algorithm on the more challenging Cordeau benchmark set, achieving an average speedup of 7.33. The performance advantage is even more pronounced than that observed on the Solomon benchmark instances. Nearly all instances require only a single run of the DP algorithm to obtain the optimal solution. It is worth noting that the model for Cordeau's instances is trained on 50-vertex random instances and evaluated on benchmark instances with 48 to 288 vertices. These results also demonstrate the effectiveness of the proposed method in generalizing to larger-scale and more complex instances. More detailed generalization performance results for different training distributions and instance sizes are presented in Appendix E. Furthermore, we observed that although the solving efficiency has improved substantially, the final size of the ng-sets has nearly tripled. This indicates that the presented algorithm still has significant room for improvement. One promising direction is to explore more advanced and better-suited learning models.

## 6 Conclusion

In this paper, we propose a novel learning-augmented exact approach, DP-NG-ML, for solving a classic combinatorial optimization problem, the Orienteering Problem with Time Windows (OPTW), which arises naturally in a wide range of transportation and logistic applications. This approach combines an advanced deep learning model with a dynamic programming (DP) algorithm enhanced by the ng-route relaxation technique. The relaxation technique relaxes the problem's elementarity restrictions, thereby effectively reducing the search space of the DP algorithm. To further improve its efficiency within a data-driven paradigm, We develop a novel Graph Convolutional Network, DiConvNet, to construct the so-called ng-sets that define this relaxation. The network is trained in a supervised manner, using optimal solutions as high-quality labels. Extensive experiments on OPTW benchmark instances demonstrate that the proposed learning-augmented exact approach outperforms the SOTA exact algorithm. Moreover, DiConvNet shows superior performance compared to widely used heuristics for constructing the sets.

Future work will explore incorporating reinforcement learning and other advanced learning models into our framework to enhance its generalization to solve large-scale problem instances with diverse data distributions. In addition, the proposed learning-augmented approach can be extended to address other related combinatorial optimization problems and potentially be integrated into column generation frameworks for solving multi-route variants.

Table 2: Comparison of our proposed algorithm against the state-of-the-art pulse algorithm on Solomon's benchmark instances.

| Instance | OptVal | pulse time(s) | DP-NG | | | DP-NG-ML | | | | $R_{pulse}$ | $R_{ml}$ |
|---|---|---|---|---|---|---|---|---|---|---|---|
| | | | ng-size | iter. | time(s) | pre-ng-size | ng-size | iter. | time(s) | | |
| c101 | 320 | 0.016 | 0 | 1 | 0.041 | 0 | 0 | 1 | 0.034 | 0.47 | **1.21** |
| c102 | 360 | 0.125 | 0.19 | 14 | 0.219 | 0.97 | 0.97 | 1 | 0.072 | **1.73** | **3.04** |
| c103 | 400 | 1.473 | 2.49 | 64 | 4.031 | 2.98 | 3.33 | 12 | 2.095 | 0.70 | **1.92** |
| c104 | 420 | 1.801 | 1.96 | 54 | 4.865 | 2.74 | 2.97 | 8 | 2.227 | 0.81 | **2.18** |
| c105 | 340 | 0.016 | 0 | 1 | 0.002 | 0 | 0 | 1 | 0.006 | **2.88** | 0.40 |
| c106 | 340 | 0.016 | 0.02 | 2 | 0.008 | 0.24 | 0.24 | 1 | 0.003 | **4.75** | **2.23** |
| c107 | 370 | 0.016 | 0 | 1 | 0.006 | 0.26 | 0.26 | 1 | 0.004 | **3.90** | **1.41** |
| c108 | 370 | 0.032 | 0.07 | 6 | 0.036 | 1.09 | 1.12 | 4 | 0.038 | 0.85 | 0.95 |
| c109 | 380 | 0.062 | 0.3 | 14 | 0.134 | 1.21 | 1.43 | 11 | 0.142 | 0.44 | 0.94 |
| r101 | 198 | 0.001 | 0 | 1 | 0.001 | 0 | 0 | 1 | 0.022 | 0.05 | 0.05 |
| r102 | 286 | 2.146 | 1.87 | 39 | 3.000 | 3.45 | 2.9 | 2 | 0.828 | **2.59** | **3.62** |
| r103 | 293 | 10.511 | 2.28 | 44 | 12.945 | 3.94 | 3.53 | 1 | 1.656 | **6.35** | **7.81** |
| r104 | 303 | 26.371 | 2.92 | 56 | 53.129 | 4.39 | 4.01 | 8 | 37.254 | 0.71 | **1.43** |
| r105 | 247 | 0.001 | 0.03 | 2 | 0.005 | 0.58 | 0.16 | 2 | 0.025 | 0.04 | 0.19 |
| r106 | 293 | 3.841 | 2.07 | 44 | 4.142 | 3.78 | 3.12 | 6 | 1.894 | **2.03** | **2.19** |
| r107 | 299 | 11.993 | 2.48 | 52 | 15.771 | 4.22 | 3.67 | 2 | 3.313 | **3.62** | **4.76** |
| r108 | 308 | 27.892 | 2.81 | 51 | 45.220 | 4.6 | 3.9 | 5 | 16.732 | **1.67** | **2.70** |
| r109 | 277 | 0.032 | 0.43 | 17 | 0.166 | 1.6 | 1.29 | 2 | 0.039 | 0.83 | **4.29** |
| r110 | 284 | 0.205 | 0.64 | 19 | 0.368 | 3.86 | 2.53 | 1 | 0.082 | **2.50** | **4.48** |
| r111 | 297 | 5.078 | 2.08 | 45 | 6.048 | 4.38 | 3.46 | 3 | 1.640 | **3.10** | **3.69** |
| r112 | 298 | 3.887 | 1.93 | 45 | 5.582 | 4.36 | 4.04 | 1 | 2.222 | **1.75** | **2.51** |
| rc101 | 219 | 0.015 | 0.03 | 3 | 0.005 | 0.33 | 0.33 | 1 | 0.035 | 0.43 | 0.14 |
| rc102 | 266 | 0.078 | 0.41 | 11 | 0.102 | 1.22 | 1.22 | 1 | 0.040 | **1.93** | **2.53** |
| rc103 | 266 | 0.235 | 1.06 | 20 | 0.418 | 1.88 | 1.88 | 1 | 0.076 | **3.07** | **5.47** |
| rc104 | 301 | 1.159 | 0.86 | 16 | 0.635 | 2.43 | 2.43 | 1 | 0.262 | **4.42** | **2.42** |
| rc105 | 244 | 0.031 | 0.33 | 13 | 0.062 | 0.89 | 0.89 | 1 | 0.010 | **3.16** | **6.27** |
| rc106 | 252 | 0.031 | 0.42 | 21 | 0.117 | 1.25 | 1.25 | 1 | 0.015 | **2.01** | **7.58** |
| rc107 | 277 | 0.173 | 0.93 | 31 | 0.497 | 2.07 | 2.08 | 2 | 0.112 | **1.54** | **4.43** |
| rc108 | 298 | 0.549 | 0.85 | 21 | 0.475 | 2.38 | 2.38 | 1 | 0.099 | **5.55** | **4.80** |
| Mean | | 3.372 | 1.02 | 24.4 | 5.449 | 2.11 | 1.91 | 2.9 | 2.448 | **1.38** | **2.23** |

Table 3: Comparison of our proposed algorithm against the state-of-the-art pulse algorithm on Cordeau benchmark instances.

| Instance | OptVal | pulse time(s) | DP-NG | | | DP-NG-ML | | | | $R_{pulse}$ | $R_{ml}$ |
|---|---|---|---|---|---|---|---|---|---|---|---|
| | | | ng-size | iter. | time(s) | pre-ng-size | ng-size | iter. | time(s) | | |
| pr01 | 308 | 0.047 | 1.60 | 11 | 0.206 | 1.9 | 2 | 2 | 0.067 | 0.70 | **3.06** |
| pr02 | 404 | 0.376 | 1.02 | 21 | 0.981 | 3 | 3 | 1 | 0.197 | **1.91** | **4.98** |
| pr03 | 394 | 0.706 | 1.15 | 30 | 1.815 | 3 | 3 | 1 | 0.196 | **3.61** | **9.27** |
| pr04 | 489 | 1.285 | 0.93 | 25 | 4.866 | 3 | 3 | 1 | 0.441 | **2.91** | **11.03** |
| pr05 | 595 | 18.857 | 1.08 | 28 | 13.925 | 3 | 3 | 2 | 3.402 | **5.54** | **4.09** |
| pr06 | 591 | 31.883 | 0.88 | 29 | 14.262 | 3 | 3 | 2 | 5.212 | **6.12** | **2.74** |
| pr07 | 298 | 0.078 | 0.97 | 15 | 0.138 | 1.89 | 1.89 | 1 | 0.020 | **3.82** | **6.74** |
| pr08 | 463 | 0.517 | 0.89 | 15 | 1.007 | 3 | 3 | 1 | 0.180 | **2.87** | **5.60** |
| pr09 | 493 | 33.514 | 1.18 | 32 | 15.620 | 3 | 3 | 1 | 3.814 | **8.79** | **4.09** |
| pr10 | 594 | 23.802 | 0.96 | 33 | 15.680 | 3 | 3 | 1 | 1.627 | **14.63** | **9.64** |
| Mean | | 11.107 | 1.06 | 23.9 | 6.850 | 2.78 | 2.79 | 1.3 | 1.516 | **7.33** | **4.52** |

# 7 Acknowledgments

This work is supported by Guangdong Basic and Applied Basic Research Foundation (Grant No. 2023A1515110228, 2024A1515140118), Hetao Shenzhen-Hong Kong Science and Technology Innovation Cooperation Zone Project (No.HZQSWS-KCCYB-2024016), National Key R&D Program of China (Grant No. 2023YFA1009302), Science and Technology Innovation Team of Shaanxi Province (2023-CX-TD-07), the Key Research and Development Program of Shaanxi (2024GH-ZDXM-48), and the National Natural Science Foundation of China (Grant 72571280, 72101264, 72431011, 72421002).

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

# A  DP-NG algorithm Details

## A.1  DP with ng-route relaxation

In the DP algorithm, a label associated with vertex $i$ represents a path starting at depot and ending at vertex $i$. The DP algorithm repeatedly extends each label to generate new labels if all the resources constraints are satisfied. To limit the exponential increase of the number of labels, a dominance test is applied between the new label and other existing labels and the dominated one will be pruned. This algorithm can be further improved by extending labels bi-directionally and matching the labels in both directions to form complete paths [30]. Since the forward and backward extensions are symmetrical to each other, we only explain the forward labels for simplicity.

Let $L(P) = (i, \tau(P), s(P), \Pi(P))$ be the forward label associated with its last visited vertex $i$, arrival time $\tau(P)$, score $s(P)$ and memory set $\Pi(P)$. Label $L(P)$ can be extended to vertex $j$ if $i_{p+1} \notin \Pi(P)$ and $\tau(P) + d_i + t_{ij} \leq w_j^2$. After the extension, a new label $L(P')$ can be obtained from the label $L(P)$ by the following operations:

$$L(P') = (j, \max\{\tau(L) + d_i + t_{ij},\ w_j^1\}, s(P) + s_j, \Pi(L) \cap N_j \cup \{j\}) \tag{9}$$

The ng-route relaxation is incorporated into the DP algorithm by permitting the generation of non-elementary labels. Specifically, a label $L(P)$ can be extended to a previously visited vertex, provided that this vertex is not included in its memory set $\Pi(P)$.

Once a new label is generated, a dominance test is performed between the new label and existing labels associated with the same vertex. Any dominated label is pruned and will no longer be extended. Given two labels $L_1$ and $L_2$ associated with the same vertex, $L_1$ dominates $L_2$ when the following conditions hold and at least one of them is strict:

$$\begin{cases} \tau(L_1) \leq \tau(L_2), \\ s(L_1) \geq s(L_2), \\ \Pi(L_1) \subseteq \Pi(L_2). \end{cases} \tag{10}$$

These conditions indicate that any further extension from label $L_2$ can also be achieved by label $L_1$, while $L_1$ still maintains a higher collected score than $L_2$. Evidently, as the size of the ng-sets increases, each label must memorize more visited vertices, resulting in a higher-dimension state space that must be explored.

The extension of forward and backward labels is terminated at the midpoint of the planning horizon $[0, T_{max}]$, and the two are matched together to form complete paths that start and end at the depot. The matching procedure between forward label $L(P_f) = (i, \tau(P_f), s(P_f), \Pi(P_f))$ and backward label $L(P_b) = (j, \tau(P_b), s(P_b), \Pi(P_b))$ is feasible only if the following conditions are satisfied:

$$\begin{cases} \tau(P_f) + d_i + t_{ij} \leq \tau(P_b) - d_j, \\ \Pi(P_f) \cap \Pi(P_b) = \emptyset. \end{cases} \tag{11}$$

After the matching procedure, the complete path with the highest collected scores is considered the optimal solution at the current iteration. If the solution is non-elementary, the ng-sets are accordingly enlarged and the DP algorithm is rerun; otherwise, the optimal solution for the primal problem is obtained.

## A.2  Outline of DP-NG algorithm

The pseudocode of the DP-NG algorithm with core procedures is provided in Algorithm 1. **N** denotes the set of ng-sets, and **R** represents the set of all labels, i.e., the union of the label sets $\mathcal{R}_i$ for all vertices $i$. For each vertex $i$, $N_i$ is initially empty. In each iteration, once a non-elementary optimal path is computed, the ng-sets are updated to forbid all the cycles that occur in the path. Unlike standard DP algorithms [30, 25], our method retains non-dominated labels after each iteration, thereby enabling their reuse in the subsequent iteration to effectively prune unpromising labels during the dominance test.

**Algorithm 1:** The Dynamic Programming with Ng-route Relaxation algorithm

---

Define $\mathbf{N} = \bigcup N_i$; $N_i \leftarrow \emptyset$, $\forall i \in V$; //Initialize ng-sets as empty sets

**Procedure** DP-NG($\mathbf{N}$,$G$):

    **Input:** pre-defined ng-sets $\mathbf{N}$, input graph $G$

    **Output:** elementary optimal path $P^*$

    $\mathbf{R} = \bigcup_{i \in V}(\mathcal{R}_i^f \bigcup \mathcal{R}_i^b)$; $\mathcal{R}_i^f \leftarrow \emptyset$, $\mathcal{R}_i^b \leftarrow \emptyset$, $\forall i \in V$; $isElem \leftarrow$ **false**; $k \leftarrow 0$;

    **while** $not\ isElem$ **do**

        $\mathbf{R} \leftarrow$ dynamicProgramming($\mathbf{R}, \mathbf{N}, G$);

        $P_k^* \leftarrow$ selectOptimalPath($\mathcal{R}_k$);

        **if** isElementaryPath($P_k^*$) **then**

            $P^* \leftarrow P_k^*$;

            $isElem \leftarrow$ **true**;

        **else**

            $\mathbf{N} \leftarrow$ updateNGSets($\mathbf{N}, P_k^*$);

            $\mathbf{R} \leftarrow$ updateLabels($\mathbf{R}, \mathbf{N}$);

            $k \leftarrow k + 1$;

    **return** $P^*$

**Procedure** dynamicProgramming($\mathbf{R}, \mathbf{N}, G$):

    Define $\mathbf{R} = \mathbf{R}^f \bigcup \mathbf{R}^b$; $\mathbf{R}^f = \bigcup_i \mathcal{R}_i^f$, $\forall i \in V$; $\mathbf{R}^b = \bigcup_i \mathcal{R}_i^b$, $\forall i \in V$;

    $\Phi \leftarrow 0$; //Initialize unchecked vertices set with depot

    **while** $\Phi! = \emptyset$ **do**

        $i \leftarrow$ pop($\Phi$);

        $\Delta\mathbf{R}^f \leftarrow$ forwardExtension($\mathcal{R}_i^f$);

        $\mathbf{R}^f \leftarrow$ forwardDominance($\Delta\mathbf{R}^f, \mathbf{R}^f$);

        $\Delta\mathbf{R}^b \leftarrow$ backwardExtension($\mathcal{R}_i^b$);

        $\mathbf{R}^b \leftarrow$ backwardDominance($\Delta\mathbf{R}^b, \mathbf{R}^b$);

    **return** matchingProcedure($\mathbf{R}^f, \mathbf{R}^b$);

**Procedure** updateNGSets($\mathbf{N}, P$):

    **forall** cycle $C = (v, \dots, v) \in \mathcal{C}(P)$ **do**

        **forall** $j \in \mathcal{S}(C) \setminus \{v\}$ **do**

            $N_j \leftarrow N_j \bigcup \{v\}$

    **return** $\mathbf{N}$

---

# B  DP-NG-ML algorithm Details

## B.1  Other components of DiConvNet model

**MLP classifier** The edge embedding $h_{ij}$ of the last DiConvNet layer is used to compute the probability that the directed edge $e_{ij}$ belongs to the ng-sets of the graph. The output of the MLP classifier is then passed through a softmax output layer to normalize the probabilities. For each edge $e_{ij}$, the probability of being assigned label 1 is expressed as:

$$f_{ij}(G; \theta) = \text{MLP}(h_{ij}^{l_{max}}), \tag{12}$$

where $l_{max}$ represents the maximum number of DiConvNet layer.

**Loss function** . As the problem size increases, the classification task becomes increasingly imbalanced toward the negative class. To address this issue, we introduce appropriate class weights to balance the contributions of positive and negative samples in the loss function. Accordingly, we minimize the weighted binary cross-entropy loss for a mini-batch of training sample $(G^k, \mathbf{l}^k)$ where $\mathbf{l}^k \in \{0, 1\}^{|E| \times 1}$ represent the binary labels of graph $G^k$. The loss function is calculated by:

$$\mathcal{L}(\mathbf{l}^k, f(G^k; \theta)) = \frac{1}{m} \sum_{k=1}^{m} \{\omega_1^k \mathbf{l}_{ij}^k log(f_{ij}(G^k; \theta)) + \omega_0^k(1 - \mathbf{l}_{ij}^k)log(1 - f_{ij}(G^k; \theta))\}, \tag{13}$$

where $\mathbf{l}_{ij}^k$ is the label of $e_{ij}$ in $\mathbf{l}^k$, $f_{ij}(G^k; \theta)$ is the corresponding probability, and $m$ denotes the batch size. For each instance $k$, we compute balanced class weights as $\omega_0^k = \frac{|E|}{2 \cdot n_0}$ and $\omega_1^k = \frac{|E|}{2 \cdot n_1}$, where $|E|$ is the total number of edges in the graph, and $n_0, n_1$ denote the number of negative and positive samples, respectively.

## B.2 ConvNet model

The only difference between the ConvNet and DiConvNet models lies in the graph convolutional layers. The rest of the architecture remains identical.

**ConvNet layers.** Let $v_i$ and $h_{ij}$ denote the vertex and edge feature vectors at layer $l$, respectively, associated with vertex $i$ and edge $e_{ij}$. We define the vertex feature and edge feature at the next layer as follows:

$$v_i^{l+1} = v_i^l + \mathrm{ReLU}(\mathrm{BN}(\theta_4^l v_i^l + \sum_{j \in \mathcal{N}_{in}(i)} \eta_{ij}^l \odot \theta_5^l v_j^l)) \text{ with } \eta_{ij}^l = \frac{\sigma(h_{ij}^l)}{\sum_{j' \in \mathcal{N}_{in}(i)} \sigma(h_{ij'}^l) + \epsilon}, \quad (14)$$

$$h_{ij}^{l+1} = h_{ij}^l + \mathrm{ReLU}(\mathrm{BN}((\theta_6^l h_{ij}^l + \theta_7^l v_i^l + \theta_8^l v_j^l)), \quad (15)$$

where $\theta_4, \theta_5, \theta_6, \theta_7, \theta_8 \in \mathbb{R}^{h \times h}$, $\sigma$ is the sigmoid function, $\epsilon$ is a small constant, $\odot$ represents the Hadamard product, ReLU is the rectified linear unit, and BN stands for batch normalization. The initial inputs to the ConvNet are defined as $v_i^0 = \alpha_i$ for each vertex and $h_{ij}^0 = \beta_{ij}$ for each edge. This network architecture is based on the Residual Gated Graph ConvNet proposed by Bresson and Laurent [5], with an extension to incorporate edge embeddings. $\eta_{ij}^l$ acts as edge gate to govern the propagation of information from edge embeddings to vertex embeddings during each convolutional layer. Evidently, the ConvNet model produces symmetric embeddings for all edges.

## B.3 Outline of DP-NG-ML algorithm

The pseudocode of the proposed DP-NG-ML algorithm is presented in Algorithm 2. The process begins with an edge reduction procedure that tags edges and produces a reduced graph. The vertex and edge initial features are then processed by the trained DiConvNet model to predict the ng-sets. Finally, the DP-NG algorithm is executed to compute the optimal solution.

---

**Algorithm 2:** The learning-augmented DP-NG-ML algorithm

---

**Procedure** `DP-NG-ML()`:
    **Input:** trained DiConvNet model $f(:; \tilde{\theta})$, input graph $G$
    **Output:** elementary optimal path $P^*$
    $G_{reduced} \leftarrow$ edgeReduction($G$);
    $\tilde{\mathbf{N}} \leftarrow$ forwardPass($f(G_{reduced}; \tilde{\theta})$);
    $P^* \leftarrow$ DP-NG($\tilde{\mathbf{N}}, G$);
    **return** $P^*$

---

## C Datasets settings

We summarize below the parameter settings used for generating training instances in our experiments.

- **Solomon's instances.** For each instance, 50%, 75% or 100% of vertices are randomly selected to receive time windows, while the time windows of remaining vertices are assigned fixed windows $[w_i^1 + t_{0i}, w_i^2 - t_{i0} - d_i]$. For each vertex $i$, the midpoint of the time window is sampled uniformly at random from the interval $(w_i^1 + t_{0i}, w_i^2 - t_{i0} - d_i)$. The half-width of each time window is drawn from a normal distribution with mean $\mu$ and standard deviation $\delta$, where $\mu$ and $\delta$ are randomly selected integers from $[5, 50]$ and $[0, 10]$, respectively, for each instance.

- **Cordeau's instances.** To generate clustered instances, we first randomly generate several centers in the $[-50, 50]^2$ square according to a continuous uniform distribution. For instances with sizes 20, 30, and 50, the number of centers is set to 2, 3, and 4, respectively. One of the centers is

randomly selected as the depot. The vertex coordinates are randomly generated with a continuous uniform distribution within $[-100, 100]^2$, and both the score and service time are random integers within the range $[1, 25]$. For each newly generated vertex, the distance $t_c$ to the closest center is calculated. If $t_c$ is less than $e^{0.05t_c}$, the vertex is accepted; otherwise, it is discarded. This process is repeated until the specified number of vertices has been generated. For the time window of each vertex, its opening time is a random integer between $[60, 480]$, and its length is a random integer between $[90, 180]$.

# D   Parameter tuning

For each instance, the learning model produces a probabilistic heatmap over the edges, indicating the likelihood of each edge being included in the ng-sets. When evaluated on large-size instances, a significant number of misclassified edges may diminish the performance gains brought by the learning model. To preserve the improvement on benchmark instances, we only select the top $\tilde{n} \cdot |V|$ edges with the highest predicted probabilities, provided the probability exceeds 0.5. Parameter tuning experiments for $\tilde{n}$ are conducted to identify the best configuration, as summarized in Tables 4 and 5. The best-performing $\tilde{n}$ values are highlighted in bold. We observed that a larger $\tilde{n}$ tend to identify more positive samples, thereby significantly reducing the number of iterations. However, this also leads to a higher number of misclassified positives, resulting in larger ng-sets. For simpler instances like Solomon's, tuning $\tilde{n}$ has limited impact on performance. However, for larger and more complex instances such as Cordeau's, selecting an appropriate $\tilde{n}$ is crucial for achieving good performance.

Table 4: Performance of our proposed approach with different $\tilde{n}$ on Solomon's benchmark instances.

| Solomon | DiConvNet | | | DP-NG-ML | | |
|---|---|---|---|---|---|---|
| $\tilde{n}$ | Error Rate(%) | FNR(%) | FPR(%) | ng-size | iter. | time(s) |
| 0.5 | 2.78 | 45.94 | 1.21 | 1.05 | 16.3 | 4.47 |
| 1 | 2.84 | 31.25 | 1.77 | 1.19 | 11.6 | 3.63 |
| 1.5 | 2.83 | 22.62 | 2.12 | 1.33 | 8.1 | 3.34 |
| 2 | 2.90 | 17.27 | 2.44 | 1.48 | 5.6 | 3.05 |
| 2.5 | 3.06 | 13.78 | 2.77 | 1.64 | 4.0 | 2.62 |
| 3 | 3.25 | 12.09 | 3.06 | 1.78 | 3.2 | 2.45 |
| 3.5 | 3.36 | 11.43 | 3.21 | 1.87 | 3.0 | 2.50 |
| **4** | **3.41** | **11.21** | **3.27** | **1.91** | **2.9** | **2.41** |
| **4.5** | **3.41** | **11.21** | **3.27** | **1.91** | **2.9** | **2.41** |
| **5** | **3.41** | **11.21** | **3.27** | **1.91** | **2.9** | **2.41** |

Table 5: Performance of our proposed approach with different $\tilde{n}$ on Cordeau's benchmark instances.

| Cordeau | DiConvNet | | | DP-NG-ML | | |
|---|---|---|---|---|---|---|
| $\tilde{n}$ | Error Rate(%) | FNR(%) | FPR(%) | ng-size | iter. | time(s) |
| 0.5 | 2.10 | 58.18 | 0.17 | 1.06 | 14.0 | 4.47 |
| 1 | 1.77 | 33.38 | 0.72 | 1.23 | 8.2 | 3.30 |
| 1.5 | 2.20 | 18.86 | 1.69 | 1.58 | 4.3 | 2.44 |
| 2 | 3.07 | 12.03 | 2.82 | 2.01 | 2.5 | 1.87 |
| 2.5 | 3.75 | 8.43 | 3.59 | 2.39 | 1.7 | 1.59 |
| **3** | **4.49** | **6.40** | **4.40** | **2.79** | **1.3** | **1.49** |
| 3.5 | 5.18 | 5.27 | 5.13 | 3.16 | 1.3 | 1.84 |
| 4 | 5.71 | 4.43 | 5.69 | 3.48 | 1.2 | 1.90 |
| 4.5 | 6.17 | 4.08 | 6.16 | 3.76 | 1.2 | 2.19 |
| 5 | 6.53 | 3.85 | 6.53 | 4.01 | 1.2 | 2.75 |

# E More experimental results

## E.1 Test on more difficult instances

The SOTA pulse algorithm was evaluated by Duque et al. [12] only on the standard Solomon and Cordeau benchmark instances, which have been tested in Tables 2 and 3. To further assess the performance of our algorithm on harder instances, we generated 20 Cordeau-style new instances with 100 vertices and time windows twice as wide. To ensure robust performance, the model was retrained using 10,000 labeled instances with 50 vertices and wide time windows. The computational times (in seconds) are reported in Table 6.

The results show that the learning-based method accelerates the DP-NG algorithm by a factor of 1.61 on average, while still outperforming the state-of-the-art pulse algorithm. This limited improvement mainly stems from prediction accuracy. Increasing both the number and size of training instances could further enhance the results.

Table 6: Comparison of our proposed algorithm against the state-of-the-art pulse algorithm on Cordeau-style instances with wide time windows.

| Instance | OptVal | SOTA-pulse time(s) | DP-NG ng-size | DP-NG Iter. | DP-NG time(s) | DP-NG-ML ng-size | DP-NG-ML Iter. | DP-NG-ML time(s) | $R_{pulse}$ | $R_{ml}$ |
|---|---|---|---|---|---|---|---|---|---|---|
| 100_0 | 511 | 71.77 | 5.56 | 67 | 367.28 | 6.99 | 12 | 281.19 | 0.26 | **1.31** |
| 100_1 | 485 | 19.29 | 5.79 | 67 | 161.33 | 6.51 | 18 | 113.25 | 0.17 | **1.42** |
| 100_2 | 498 | 18.51 | 3.55 | 38 | 35.85 | 5.92 | 2 | 26.15 | 0.71 | **1.37** |
| 100_3 | 482 | 13.66 | 2.76 | 33 | 13.53 | 4.98 | 4 | 8.39 | **1.63** | **1.61** |
| 100_4 | 503 | 265.44 | 4.84 | 53 | 93.12 | 6.85 | 12 | 112.46 | **2.36** | 0.83 |
| 100_5 | 535 | 330.85 | 5.22 | 49 | 256.83 | 6.49 | 2 | 137.34 | **2.41** | **1.87** |
| 100_6 | 516 | 8.17 | 2.7 | 25 | 11.78 | 5.6 | 1 | 6.86 | **1.19** | **1.72** |
| 100_7 | 435 | 7.18 | 3.72 | 45 | 27.26 | 5.05 | 8 | 32.86 | 0.22 | 0.83 |
| 100_8 | 495 | 24.29 | 4.91 | 58 | 148.31 | 6.25 | 3 | 33.03 | 0.74 | **4.49** |
| 100_9 | 460 | 4.75 | 2.95 | 35 | 13.47 | 4.52 | 6 | 4.85 | 0.98 | **2.78** |
| 100_10 | 577 | 1022.04 | 7.35 | 83 | 1452.04 | 8.49 | 16 | 1276.55 | 0.80 | **1.14** |
| 100_11 | 538 | 10.31 | 4.14 | 54 | 92.12 | 6.69 | 6 | 86.07 | 0.12 | **1.07** |
| 100_12 | 486 | 7.60 | 5.02 | 51 | 40.95 | 6.63 | 11 | 45.22 | 0.17 | 0.91 |
| 100_13 | 482 | 46.58 | 4.12 | 49 | 66.83 | 5.2 | 7 | 38.31 | **1.22** | **1.74** |
| 100_14 | 583 | 1031.12 | 4.14 | 39 | 97.24 | 7.45 | 2 | 129.05 | **7.99** | 0.75 |
| 100_15 | 493 | 89.88 | 4.2 | 60 | 192.83 | 6.21 | 5 | 235.07 | 0.38 | 0.82 |
| 100_16 | 462 | 6.90 | 3.3 | 34 | 10.92 | 4.79 | 7 | 6.14 | **1.12** | **1.78** |
| 100_17 | 505 | 175.96 | 5.81 | 66 | 695.98 | 6.41 | 16 | 450.45 | 0.39 | **1.55** |
| 100_18 | 488 | 2293.26 | 6.54 | 56 | 363.42 | 6.82 | 6 | 149.93 | **15.30** | **2.42** |
| 100_19 | 463 | 31.56 | 4.15 | 56 | 38.37 | 5.01 | 16 | 20.67 | **1.53** | **1.86** |
| Mean | | 273.96 | 4.54 | 50.9 | 208.97 | 6.14 | 8.0 | 159.69 | **1.98** | **1.61** |

## E.2 Comparison with DiDPPy solver

We tested a generic DP solver, DiDPPy [20], on the OPTW using Cordeau instances. For a fair comparison, we added forced transitions that treat unvisited vertices as visited if they are infeasible due to time window constraints at the current state. In this case, more unpromising states will be pruned. Unfortunately, instances with more than 90 vertices ran out of memory using DiDPPy. We select the first 90 vertices (excluding the depot) from each instance to create new instances. The computational time in seconds are shown in Table 7. The number of vertices is indicated by the suffix in the name. The results show that the DP-NG algorithm is 100 times faster than the DiDPPy using the CABS solver. With the learning method integrated, the speedup reaches up to 800 times.

# F Generalization performance

Figure 5 provides the results of the proposed DP-NG-ML algorithm with various training dataset distributions of Cordeau's instances and instance sizes. The clustered training datasets are denoted as "Train-C|V|", where $|V|$ is the instance size. Instances with uniform distributions are labeled as "R". The parameter $\tilde{n}$ is set to 3 in all experiments. Overall, on smaller instances, the proposed method performs better when trained on smaller-sized instances. However, for generalization to larger instances, training the model on larger-sized instances tends to yield better performance. This is because larger instances offer greater potential for performance improvement through learning-augmented approach, and models trained on larger data are more capable of identifying a greater number of

Table 7: Comparison of our proposed algorithm against the generic dynamic programming solver DiDPPy.

| Instance | OptVal | DiDPPy | DP-NG | Speedup1 | DP-NG-ML | Speedup2 |
|----------|--------|--------|-------|----------|----------|----------|
| pr01_48 | 308 | 3.094 | 0.233 | 13.29 | 0.079 | 38.93 |
| pr02_90 | 392 | 148.266 | 0.882 | 168.03 | 0.139 | 1065.18 |
| pr03_90 | 331 | 51.841 | 0.782 | 66.29 | 0.231 | 224.82 |
| pr04_90 | 356 | 33.328 | 0.674 | 49.43 | 0.169 | 197.45 |
| pr05_90 | 443 | 193.404 | 1.386 | 139.54 | 0.551 | 350.70 |
| pr06_90 | 329 | 29.954 | 0.527 | 56.85 | 0.046 | 656.43 |
| pr07_72 | 298 | 12.956 | 0.187 | 69.14 | 0.016 | 810.99 |
| pr08_90 | 374 | 19.59 | 0.397 | 49.30 | 0.034 | 568.74 |
| pr09_90 | 320 | 100.259 | 0.346 | 289.93 | 0.057 | 1757.46 |
| pr10_90 | 399 | 169.158 | 0.862 | 196.35 | 0.069 | 2465.47 |
| Mean | | 76.185 | 0.628 | 109.81 | 0.139 | 813.62 |

positive samples. Moreover, even when the coordinate distribution of the training instances differs from that of the test instances, the method still demonstrates significant performance improvements.

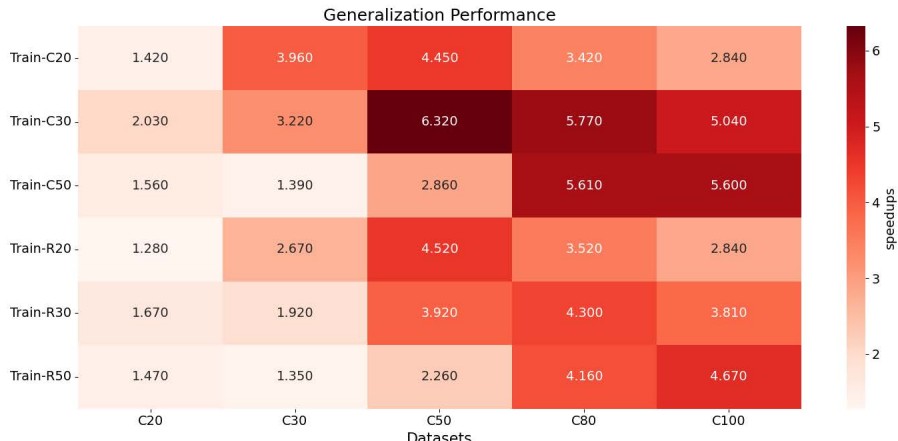

Figure 5: Evaluation of DP-NG-ML algorithm performance on Cordeau's instances with varying sizes and training distributions. The vertical axis represents training instances with clustered and uniform distributions of sizes 20, 30, and 50. The horizontal axis shows test instances with different sizes under clustered distribution. The heatmap illustrates the speedups achieved under various configurations relative to the standard DP-NG algorithm.

## G  Visualizing Model Predictions

Figures 6 display the visualizations of ground truth and predicted ng-sets for Cordeau's benchmark "pr01" with 48 vertices. The comparative visualizations demonstrate that the proposed learning models effectively capture the structural patterns of ng-sets, successfully identifying a large proportion of positive samples from a very large sample pool. The edge set in the ground-truth labels exhibits certain structural properties, such as short distances between nodes and proximity to the depot node. These properties, to some extent, help the neural network better identify and predict. The comparison between Figures 6 (b) and (c) shows that the DiConvNet model can more effectively capture the asymmetry of edges than the ConvNet model.

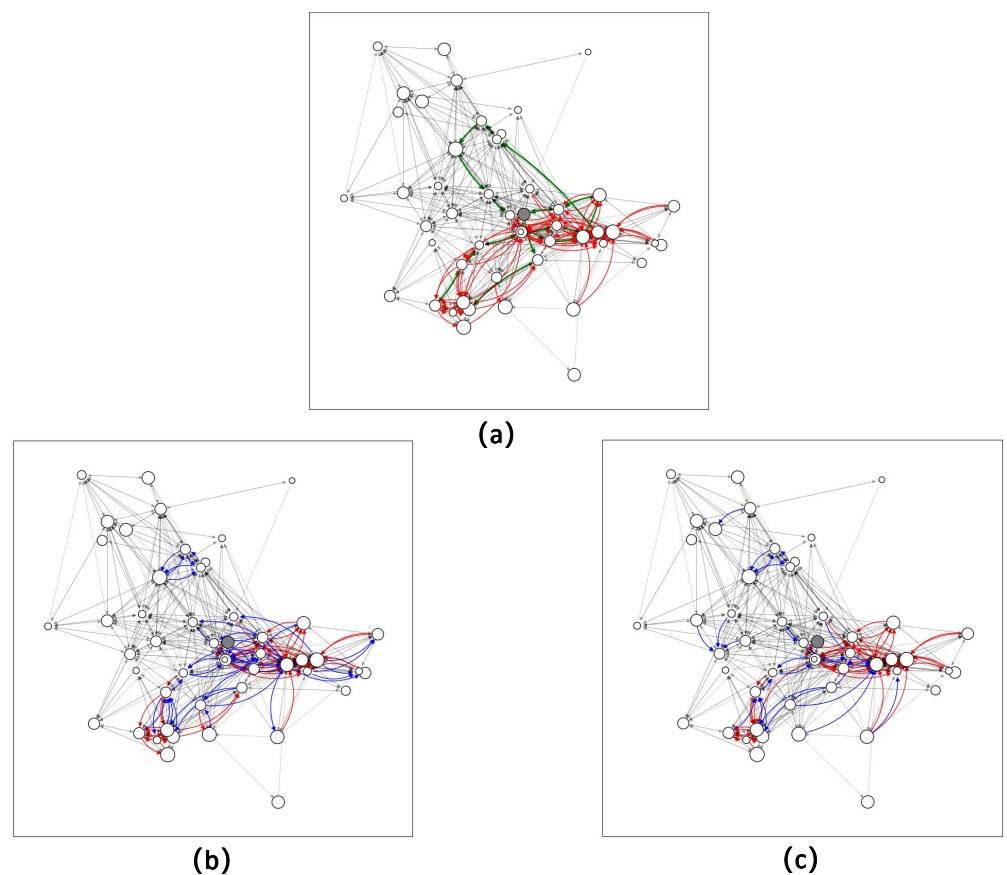

Figure 6: Visualizations of (a) the ground truth ng-sets, and the ng-sets predicted by (b) ConvNet and (c) DiConvNet. Red arrows represent positive samples, while blue arrows indicate false positives. In (a), the tour shown with green arrows denotes the optimal path. Circle sizes are scaled proportionally to the vertex scores.

