# OpenReview forum: "A Learning-Augmented Dynamic Programming Approach for Orienteering Problem with Time Windows"
_NeurIPS.cc/2025/Conference — NeurIPS 2025 poster_

### Official Review · Reviewer_7zHH · 2025-06-28

**Clarity:** 3
**Significance:** 2
**Originality:** 3
**Rating:** 5
**Confidence:** 3

**Summary:**

The paper presents a learning-augmented exact algorithm for the Orienteering Problem with Time Windows (OPTW) which is a combinatorial optimization problem (COP). The authors first reduce the number of likely edges in the ng-set from all the possible edges, then use a Graph Convolutional Network (GCN) to estimate the probability that each remaining edge is in the ng-set. The edges with the highest probabilities are then selected to initialize the ng-set and the DP-NG algorithm is used to compute the exact optimal solution. The authors test their algorithm empirically and show that it produces better ng-set initializations than a ConvNet and some heuristics. On OPTW benchmarks their algorithm produces significant speed-ups when compared to the current SOTA pulse algorithm.

**Questions:**

- Sections 3.2 and 3.3 should be rewritten to better explain the proposed algorithm (see weaknesses). This would greatly help readers which are less familiar with the underlying concepts understand the proposed algorithm without needing to refer to external ressources.
- The edge reduction part should be briefly described in the text. When that part is introduced at the beginning of Section 4.1, the authors reference the appendix without further explanations but later in the main text they mention concepts presented in the appendix.

**Ethical Concerns:**

["NO or VERY MINOR ethics concerns only"]

**Final Justification:**

After reading the other reviews and the authors’ rebuttal, I’ve decided to keep my score at 5 (Accept). Although the authors addressed my main concerns, I’m not comfortable raising to a 6 given the additional issues noted by other reviewers and my lingering uncertainty about how this work fits into the existing literature and state of the art.

**Limitations:**

Yes

**Paper Formatting Concerns:**

I have no paper formatting concerns.

**Quality:**

3

**Strengths And Weaknesses:**

### Strengths
- The presented algorithm is innovative and performs strongly on benchmarks.
- The paper is generally well written except for sections 3.1 and 3.2 (see weaknesses).
- The figures significantly help understanding the proposed method.

### Weaknesses
- While I think the paper is generally well written, Section 3.2 is somewhat confusing. I was only able to properly grasp the explained concepts using outside ressources. I think that section should be re-written and that Appendix A.1 should be incorporated into the main text since the math there helps understand things. More generally the interaction between the DP algorithm and the ng-route relaxation should be better explained; when talking about "extending a path P" the authors give the impression that the path is built vertex by vertex with the ng-set updated after every new addition. However in Figure 2 it is clearly shown that at every iteration of the DP-NG algorithm a full "optimal path" is proposed and the ng-set is then updated. Perhaps a part of this confusion is a result of my relative unfamiliarity with the exact concepts of interest, but the text and explanations should still be improved and made clearer so that the proposed solution is easier to understand. The figures help in this regard so perhaps a better alignment between the text and the figures would be useful.

---

> ### Author Rebuttal · Authors · 2025-07-26
>
> We sincerely appreciate your thorough review and valuable comments. We have re-evaluated the text in Section 3 and will revise it accordingly. In addition, a brief introduction to the ``edge reduction'' will be included in Section 4. We summarize the modifications in the following four parts and review our contributions:
>
> **1. Dynamic programming formulation**
>
> We will revise the last paragraph of Section 3.1 as follows, providing a clearer explanation of the dynamic programming formulation.
>
> _"The OPTW can be formulated as a special case of the classic Resource constrained Elementary Shortest Path Problem [1], for which no polynomial-time algorithm is known. Dynamic programming is considered an efficient exact approach for solving this problem. Let $F(S, \tau, i)$ denote the maximum collected score of a path starts from the depot and ends at vertex $i$, visiting each vertex in set $S$ exactly once, with an elapsed time of $\tau$. $F(S, \tau, i)$ can be computed by solving the recurrence equation:_
>
> \begin{equation}F(S, \tau, i) = \max_{(j,i)\in E}\\{F_i(S\setminus \\{i\\}, \tau^{\prime}, j) + s_j | \tau+t_{ji}\leq w_i^2, w_j^1 \leq \tau^{\prime}\leq w_j^2\\}, \forall i\in V, S\subseteq V,   w_i^1 \leq \tau \leq w_i^2.  \quad \(1\)\end{equation}
>
>  _Set $S$ denotes the set of visited vertices and can be interpreted as a set of dummy resources, each with a unit capacity. This set is typically encoded as a binary vector of length $|V|$. Clearly, the size of the state-space graph is heavily influenced by this vector which enforces the elementarity restrictions over all the vertices in $V$.To reduce the number of states to be explored, we employ the ng-route relaxation technique Baldacci et al. [2] to project the original state space $(S, \tau, i)$ onto a lower-dimensional space, inspired by the state space relaxation procedure proposed by Christofides et al. [3] and developed by  Boland et al. [4], Righini and Salani [1]."_
>
> **2. Clearer explanation of ng-route relaxation**
>
> We rephrase the entire Section 3.2 to better clarify the concept of ng-route relaxation as follows:
>
> _"The ng-route relaxation, introduced by  Baldacci et al. [2], provides a good compromise between enforcing elementarity constraints and enabling efficient exploration of the state space. For each vertex $i \in V$, we define an ng-set $N_i  \subseteq V$,  which is a selected subset of vertices associated with vertex $i$ (according to some criterion). Let $P=\\{0,i_1,...,i_p\\}$ be a partial path starting from the depot to vertex $i_p$, which is associated with the set of visited vertices $\mathcal{S}(P)$ and an elapsed time $\tau$. The basic DP algorithm explores the state space by extending paths to all possible succeeding vertices using equation (1). We define $\Pi(P) \subseteq \mathcal{S}(P)$ as the `memory' of path $P$. During the past extension of path $P$, for any intermediate vertex $i_k$, only the visited vertices that belong to the ng-set of $i_k$ can be retained in memory. Therefore, the memory set $\Pi(P)$ is defined as the intersection of the visited vertices and the ng-sets of all subsequently visited vertices, and is given by:_
> \begin{equation}
> 	\Pi(P)=\\{i_k\in \mathcal{S}(P)\setminus \\{i_{p}\\}:i_k\in \cap_{q=k+1}^{p} N_{i_q}\\}\cup \\{i_{p}\\}.   \quad \(2\)
> \end{equation}
> _In this way, the original state $(\mathcal{S}(P), \tau, i)$ of path $P$ is projected onto a lower-dimensional state $(\Pi(P), \tau, i)$, where $\Pi(P)$ can be encoded as a binary vector of length $|N_i|$. Since $|N_i|\leq |V|$ for each vertex $i$, the size of the state-space graph is significantly reduced. Figure 1 illustrates the path extension using given ng-sets. Note that if a visited vertex is excluded from the memory of path $P$, it may be revisited in the future extensions of $P$. This is the main drawback of the relaxation technique. As a result, the search in the relaxed state space does not guarantee to find optimal solution. A practical compromise is to iteratively tighten the relaxation, i.e., enlarge the ng-sets, based on the optimal non-elementary path obtained in each iteration of DP, until an optimal elementary path is found._
>
> _Note that this iterative framework may incur higher computational costs compared to the basic DP algorithm, as the dimensionality of the state space increases with each iteration. An inappropriate criterion for expanding the ng-sets may introduce `unnecessary' vertices, further inflating the search space. We demonstrate that this issue can be mitigated by leveraging machine learning techniques to predict the ng-sets based on hidden patterns extracted from data. Moreover, the data-driven approach serves as an effective heuristic to initialize the ng-sets, thereby reducing the number of required iterations. Essentially, our goal is to train a neural network that approximates the projection function, mapping the original state space to a suitable lower-dimension space in a one-shot manner. In this case, the ng-set of vertex $i$ can be viewed as a set of directed edges originating from $i$."_
>
> **3. A concise version of DP with ng-route relaxation**
>
> We move part of the content from Section 3.3 to Section 3.2, and rephrase this Section accordingly:
>
> _"A state of path $P$ can be interpreted as a label $L(P)=(i,\tau(P),s(P),\Pi(P))$, where $i$ is the last visited vertex and $s(P)$ is the collected score. Extending label $L(P)$ to vertex $j$ indicates adding $j$ to the end of $P$, thereby generating a new label. This extension is feasible only if the time window constraint at $j$ is satisfied and $j$ has not been visited by path $P$, i.e., $j \notin \Pi(P)$. To limit the exponential growth in the number of labels, a dominance test is applied to identify and discard labels that cannot lead to an optimal solution. By exploring the state space defined by the current ng-sets, the path with the highest scores is considered the optimal path for that iteration. If the obtained path is non-elementary, the vertices involved in a cycle is updated to include the duplicate vertex, thereby preventing the same cycle from occurring in the next iteration. This exact approach is referred to as the DP-NG algorithm, and a detailed explanation is provided in Appendix A.2. Figure 2 illustrates an example of the DP-NG algorithm applied to a seven-vertex instance. It can be expected that states with lower time consumption and higher collected scores are more likely to survive the dominance test. Consequently, pairs of vertices with high individual scores, lose spatial proximity and highly overleaped time windows are more likely to form cycles. This observation motivates the use of graph structural features for predicting the ng-sets."_
>
> **4. The content of edge reduction procedure**
>
> We move the content of the edge reduction procedure from  Appendix B.1 to the main text, and rephrase it as follows:
>
> _"To improve prediction accuracy, we propose an edge reduction procedure that leverages problem-specific characteristics to exclude edges that cannot belong to ng-sets. Two types of edges are excluded: (1) the edges connected to the depot or self-loops; (2) any edge $e\_{ij}$ satisfying the following condition:_
>
> \begin{equation}
> \max \\{w_j^1 + d_j + t_{ji}, w_i^1\\} + d_i + t_{ij} > w_j^2.   \quad \(3\)
> \end{equation}
> _The first type of edges is excluded by definition, while the second type corresponds to a situation where cycle $(j-i-j)$ is infeasible due to the time window constraints."_
>
> **5. Review of our contributions**
>
> We sincerely appreciate the constructive suggestion on reorganizing the content of Section 3. This provides a valuable opportunity for us to reflect more deeply on the innovations of our methodology and its potential contributions to both the OR and AI communities.
>
> In essence, our work successfully demonstrates a novel paradigm that integrates machine learning (ML) technique with dynamic programming for solving combinatorial optimization problems (COPs), particularly in the context of network optimization. A variety of such NP-hard problems can be solved using dynamic programming, but the approach often suffers from the 'curse of dimensionality'. Approximating the state-space relaxation with ML enables dynamic programming to operate in a lower-dimensional state space, rather than the original one, while preserving optimality. In our case, we focus on the elementarity restrictions over all the vertices in $V$, which correspond to $|V|$ state variables in dynamic programming recursion. With accuracy prediction of the state space relaxation, the number of state variables can be reduced to $|N_i|$, where $N_i \subseteq V$.
>
> According to the recent survey on ML for COPs by Bengio et al. [5], no prior research has presented a similar paradigm. We believe that this work serves as a valuable starting point and demonstrates the potential of our approach to be generalized to other network optimization problems.
>
> **Reference**
>
> [1] G. Righini and M. Salani. Decremental state space relaxation strategies and initialization heuristics for solving the orienteering problem with time windows with dynamic programming. Computers & Operations Research, 36(4):1191–1203, 2009.
>
> [2] R. Baldacci, A. Mingozzi, and R. Roberti. New Route Relaxation and Pricing Strategies for the Vehicle Routing Problem. Operations Research, 59(5):1269–1283, 2011.
>
> [3] N. Christofides, A. Mingozzi, and P. Toth. State-space relaxation procedures for the computation of bounds to routing problems. Networks, 11(2):145–164.
>
> [4] N. Boland, J. Dethridge, and I. Dumitrescu. Accelerated label setting algorithms for the elementary resource constrained shortest path problem. Operations Research Letters, 34(1):58–68, Jan. 2006.
>
> [5] Y. Bengio, A. Lodi, and A. Prouvost. Machine learning for combinatorial optimization: A methodological tour d’horizon. European Journal of Operational Research, 290(2):405–421, 2021.

---

> > ### Comment · Reviewer_7zHH · 2025-08-01
> >
> > Thank you for addressing my concerns about the clarity of the text. After reading the other reviewers’ comments and the authors’ responses, I have decided to keep my score at 5 (Accept).

---

### Official Review · Reviewer_AJaq · 2025-06-30

**Clarity:** 3
**Significance:** 2
**Originality:** 3
**Rating:** 4
**Confidence:** 2

**Summary:**

The paper studies the problem of solving NP-hard combinatorial optimization problem using machine-learned algorithms. Specifically, it considers the Orienteering Problem with Time Windows, which is essentially a variant of prize-collecting TSP equipped with time windows. The authors chose this problem to demonstrate the power of their approach compared to previously studied learning-augmented algorithms for combinatorial optimization problems with less dimensions, such as plain TSP. Problems with that many constraints and dimension have not been studied much before using such techniques.
The main contributions can be summarized as follows:

- The authors present a first approach to integrate a graph neural network with an exact dynamic programming solver for combinatorial optimization problems, accelerating exact optimization without sacrificing optimality.

- They demonstrate substantial speedups (up to 7x on challenging benchmarks) over the state-of-the-art pulse algorithm on standard OPTW instances, which highlights both efficiency gains and practical applicability of their approach.

**Questions:**

Have you considered different learning techniques that do not rely on supervised learning?

**Ethical Concerns:**

["NO or VERY MINOR ethics concerns only"]

**Final Justification:**

The authors cleared out some of my concerns regarding the applicability of their result. Since I am not an expert, some doubts still remain. Hence, I would like to keep my overall positive impression. However, I feel not confident enough to strongly support the papers acceptance.

**Limitations:**

As far as I can see, the authors have addressed major limitations.

**Paper Formatting Concerns:**

The paper formatting looks good.

**Quality:**

3

**Strengths And Weaknesses:**

### Strengths

- The authors combine learning with an exact DP method, which has several benefits. I think this is relatively rare and an important contribution to ML for combinatorial optimization.

- The authors tackle quite complex combinatorial optimization problems

- The empirical results look impressive, with quite strong speedups over previous work and on competitive benchmarks.

- The methodology seems to be generalizable and extendable to other combinatorial optmization problems.

### Weaknesses

- The method is based on supervised learning, and thus, requires labels of optimal solutions for instances. This is a significant limitation, as larger instances cannot be included in the learning process.

---

> ### Author Rebuttal · Authors · 2025-07-27
>
> Thanks for your valuable comments and suggestions. We understand the limitations of supervised learning; however, due to the problem-specific characteristics, it remains a suitable and effective choice in our context.
>
> **1. Brief review of the proposed method.**
>
> To better explain the reasons, we rewrite the dynamic programming formulation of the OPTW as follows.
>
> _Let $F(S, \tau, i)$ denote the maximum collected score of the path a path starts from the depot and ends at vertex $i$, visiting each vertex in set $S$ exactly once, with an elapsed time of $\tau$. $F(S, \tau, i)$ can be computed by solving the recurrence equation_
> \begin{equation}F(S, \tau, i) = \max_{(j,i)\in E}\\{F_i(S\setminus \\{i\\}, \tau^{\prime}, j) + s_j | \tau+t_{ji}\leq w_i^2, w_j^1 \leq \tau^{\prime}\leq w_j^2\\}, \forall i\in V, S\subseteq V,   w_i^1 \leq \tau \leq w_i^2.  \quad \(1\)\end{equation}
> _Set $S$ denotes the set of visited vertices and can be interpreted as a set of dummy resources, each with a unit capacity. This set is typically encoded as a binary vector of length $|V|$. Clearly, the size of the state-space graph is heavily influenced by this vector which enforces the elementarity restrictions over all the vertices in $V$._
>
> _For each vertex $i \in V$, we define an ng-set $N_i \subseteq V$,  which is a selected subset of vertices associated with vertex $i$ (according to some criterion). Let $P=\\{0,i_1,...,i_p\\}$ be a partial path starting from the depot to vertex $i_p$, which is associated with the set of visited vertices $\mathcal{S}(P)$ and an elapsed time $\tau$. We define $\Pi(P) \subseteq \mathcal{S}(P)$ as the ``memory'' of path $P$. During the past extension of path $P$, for any intermediate vertex $i_k$, only the visited vertices that belong to the ng-set of $i_k$ can be retained in memory. Therefore, the memory set $\Pi(P)$ is defined as the intersection of the visited vertices and the ng-sets of all subsequently visited vertices, and is given by:_
> \begin{equation}
> 	\Pi(P)=\\{i_k\in \mathcal{S}(P)\setminus \\{i_{p}\\}:i_k\in \cap_{q=k+1}^{p} N_{i_q}\\}\cup \\{i_{p}\\}.   \quad \(2\)
> \end{equation}
> _In this way, the original state $(\mathcal{S}(P), \tau, i)$ of path $P$ is projected onto a lower-dimensional state $(\Pi(P), \tau, i)$, where $\Pi(P)$ can be encoded as a binary vector of length $|N_i|$. Since $|N_i|\leq |V|$ for each vertex $i$, the size of the state-space graph is significantly reduced. The main drawback of such relaxation technique is that some original state corresponding to an infeasible solution may be projected onto a state corresponding to a feasible solution in the lower-dimensional space. As a result, the search in the relaxed state space does not guarantee to find optimal solution. A practical compromise is to iteratively tighten the relaxation, i.e., enlarge the ng-sets, based on the optimal non-elementary path obtained in each iteration of DP, until an optimal elementary path is eventually found._
>
> The central idea of this work is to train a neural network that approximates the projection function, mapping the original state space to a suitable lower-dimensional space. This enables dynamic programming to operate in a reduced state space while preserving optimality.
>
> **2. Difficulties of implementing unsupervised learning.**
>
> We have attempted to implement such an approximation using unsupervised learning framework such as reinforcement learning. However, the main challenge lies in evaluating the quality of states and actions.
>
> Existing research on solving combinatorial optimization problems (COPs) using machine learning primarily focuses on constructing solutions incrementally or directly searching for solutions based on probabilistic heatmaps. It is straightforward to evaluate the quality of any states (incumbent solution) or actions (an local movement) based on the given objective function. However, in our case, the goal is not to approximate a feasible solution to the OPTW. Instead, we aim to predict the smallest possible ng-set for each vertex such that a single run of the DP-NG algorithm can compute the optimal elementary path under the given ng-sets. Consequently, there are two challenges to implement an unsupervised learning:
>
> **First**, the delayed reward of predicting any edges in ng-sets, i.e. the size of final ng-sets, are only revealed once the optimal solution of DP-NG algorithm is elementary. This procedure often requires many steps of prediction, as well as many iterations of the DP-NG. No immediate reward signals are available when selecting a directed edge. Moreover, each run of the DP-NG to confirm the status of ng-sets is essentially equivalent to solving an NP-hard problem, despite the reduced dimensionality of the state space. According to the work of Boland et al. [1], the worst-case complexity of the basic DP algorithm without relaxation is $\mathcal{O}(|E| (T_{max} + 1)2^{|V|})$. Similarly, the worst-case complexity of te DP-NG algorithm can be approximated as $\mathcal{O}(|E| (T_{max} + 1)2^{|\bar{N}|})$, where $|\bar{N}|$ denotes an averaged size of the ng-sets across all vertices.
>
> **Second**, the final ng-sets are obtained by iteratively tighten the relaxation based on the computed optimal non-elementary paths. Specifically, when a non-elementary path is computed, for each loop that appears in the path, the duplicate vertex is inserted into the ng-sets of the vertices involved in that loop. In this way, the same loops are prevented in the next iteration of the DP-NG algorithm, based on the extension method described in Section 3.2. The inclusion of other candidate directed edges into the ng-sets does not affect the optimal non-elementary path. If an auto-regressive learning approach is adopted, many of the trials may be incorrect, and no direct feedback signals are available. Moreover, it requires running the DP-NG algorithm at each step to evaluate whether the optimal path has changed and to determine whether the prediction process should terminate. More importantly, the cost of trial-and-error is prohibitively high, as the computational complexity of the DP-NG algorithm grows exponentially with the size of the ng-sets.
>
> **3. Efforts in supervised learning.**
>
> For the above reasons, collecting a large amount of exploratory data with reward signals is extremely time-consuming. Compared to unsupervised learning, supervised learning typically requires a smaller amount of (high-quality) training data. Moreover, initializing the ng-sets to reduce the number of DP-NG iterations is inherently a one-shot process, which naturally aligns with the supervised learning paradigm.
>
> To efficiently obtain a large number of labeled samples, we enhanced the DP-NG algorithm by reusing non-dominated labels from the previous iterations. To ensure optimality, a minor modification to the original DP-NG algorithm is necessary, although we omit the formal proof in this paper. The source code of the improved DP-NG algorithm will be made publicly available after submission. The computational time (in seconds) on Cordeau benchmark instances are provide as follows:
>
> | Instance | SOTA pulse | original DP-NG | imporved DP-NG |
> |----------|------------|----------------|----------------|
> | pr01     | 0.047      | 0.237          | 0.206          |
> | pr02     | 0.376      | 1.059          | 0.981          |
> | pr03     | 0.706      | 3.424          | 1.815          |
> | pr04     | 1.285      | 7.122          | 4.866          |
> | pr05     | 18.857     | 29.005         | 13.925         |
> | pr06     | 31.883     | 30.129         | 14.262         |
> | pr07     | 0.078      | 0.203          | 0.138          |
> | pr08     | 0.517      | 1.833          | 1.007          |
> | pr09     | 33.514     | 26.698         | 15.620         |
> | pr10     | 23.802     | 30.446         | 15.680         |
> | Mean     | 11.107     | 13.016         | 6.850          |
>
> The improved DP-NG algorithm runs twice as fast as the original version and even outperforms the state-of-the-art Pulse algorithm on these instances.
>
> **4. Positioning of this Work**
>
> The main contribution of this work is to improve an exact algorithm with machine learning technique for an NP-hard COP. We successfully demonstrates a novel paradigm that approximates the state space relaxation to enable dynamic programming to operate in a lower-dimensional state space. This method shows great potential for generalization to a broad class of network optimization problems, as many of these problems can be formulated using dynamic programming recursions.
>
> **Reference**
>
> [1] N. Boland, J. Dethridge, and I. Dumitrescu. Accelerated label setting algorithms for the elementary resource constrained shortest path problem. Operations Research Letters, 34(1):58–68, 2006.

---

> > ### Comment · Reviewer_AJaq · 2025-08-04
> >
> > I thank the authors for clarifying their choice of supervised learning. After reading the other reviews, I would like to keep my overall positive impression. I still have slight concerns regarding the applicability of the method for real world instances.
> >
> > Since I am not an expert in this field, I have the feeling that I am fine if the paper gets into NeurIPS, but I cannot find good arguments to strongly support its acceptance.

---

### Official Review · Reviewer_Ax3s · 2025-07-03

**Clarity:** 4
**Significance:** 2
**Originality:** 2
**Rating:** 4
**Confidence:** 5

**Summary:**

This paper introduces a learning-augmented exact algorithm, DP-NG-ML, to solve the NP-hard Orienteering Problem with Time Windows (OPTW), aiming to maximize collected scores by visiting vertices within specified timeframes. Unlike traditional exact algorithms, this approach leverages a deep learning model called DiConvNet, a specialized graph convolutional network, to learn effective relaxations of problem restrictions from data. To enhance prediction accuracy, an initial edge reduction procedure is applied to tag and remove edges unlikely to be part of any "ng-set" before the data is fed into the DiConvNet model. DiConvNet predicts directed edges that define these relaxations, enabling significant performance gains in an exact dynamic programming algorithm. Experimental results show that this learning-augmented algorithm surpasses state-of-the-art exact algorithms, achieving a 38% speedup on Solomon's benchmark and a more than sevenfold improvement on Cordeau's benchmark, demonstrating its effectiveness in combining machine learning with operations research to preserve optimality while enhancing performance.

**Questions:**

- What is the specific methodology for edge reduction? Is this process machine learning-based, and if so, what approach is used? Additionally, is the same edge reduction technique applied consistently across all baseline methods to ensure fair comparison?

- Has any sample complexity analysis been conducted to determine the minimum number of training instances required for effective model performance? Understanding the data requirements would provide valuable insights into the practical applicability of the approach.

**Ethical Concerns:**

["NO or VERY MINOR ethics concerns only"]

**Final Justification:**

I have updated my score based on the newly added experimental results. First, the comparison to DIDPPy provides a nice baseline to one of the easy-to-use generic DP solver. Second, the comparison on larger/harder instances underscores the usefulness of such an approach. I would highly encourage adding these two results to the main text. Furthermore, I would advise the authors to add sample complexity results to the appendix. I would have preferred if the evaluation was done of two or more problem classes, nevertheless, I feel that the current contribution is impactful enough to be accepted at the conference.

**Limitations:**

- Add different and challenging problem classes to the analysis to make the contribution more significant. For example, consider incorporating problems such as vehicle routing with time windows, job shop scheduling, graph coloring, or maximum satisfiability problems.

**Paper Formatting Concerns:**

No.

**Quality:**

2

**Strengths And Weaknesses:**

Strengths

1. The manuscript is well-written with excellent clarity and logical organization.

2. The proposed methodology demonstrates strong potential for broader applicability beyond the specific problem class presented. This suggests the approach may have significant impact across multiple application areas, which is valuable for the broader research community. For example, there are many variants of TSPTW where this can be applied.

Weaknesses

1. Evaluation focuses on only one problem class, which is insufficient to establish the generalizability of the methodology, which seems to be the main contribution of the paper. The authors should test on additional problem domains such as vehicle routing, scheduling, or other combinatorial optimization problems.

2. If exact methods solve instances in <11 seconds on average, the problems may be too trivial to demonstrate meaningful performance advantages. Similar to TSPTW where makespan/travel time objectives have different difficulty levels (I don't recall which one was more difficult), the authors should modify the objective function or constraints to create more challenging instances.

3. The comparison should probably add a DiDPPy baseline [1], which has achieved state-of-the-art results on some DP problems. It has an easy to use Python package (https://didppy.readthedocs.io/en/stable/).

References
[1] Kuroiwa, Ryo, and J. Christopher Beck. "Domain-independent dynamic programming: Generic state space search for combinatorial optimization." Proceedings of the International Conference on Automated Planning and Scheduling. Vol. 33. 2023.

---

> ### Author Rebuttal · Authors · 2025-07-28
>
> We sincerely appreciate your thorough review and constructive feedback on our manuscript. Below, we address the identified weaknesses and respond to your questions.
>
> **1. Generalization to other problems.**
>
> In general, most heuristics are more easily generalized to other optimization problems, as they do not rely on exact mathematical formulations. In contrast, efficient exact algorithms are often tailored to problem-specific characteristics, making them difficult to generalize to problems with different structures. Striking a balance between the efficiency of finding optimal solutions and the generalizability remains a significant challenge. Commercial solvers like Gurobi use generic branch-and-cut to solve various MIPs, but often overlook problem structures, making them less competitive than SOTA algorithms.
>
> This work aims to develop an efficient exact algorithm for the OPTW that surpasses a highly competitive SOTA exact method, rather than a generic yet suboptimal solver. The core of this approach is the relaxation of elementarity constraints, which is problem-specific and hard to generalize across COPs. The OPTW can be reformulated as the Resource Constrained Elementary Shortest Path Problem (RCESPP) [1], where the objective is to find an elementary path minimizing the total collected weights on vertices or edges. We compare the TSPTW with the RCESPP to further illustrate this point.
>
> We formulate the RCESPP with DP recursion. Let $F(S, \tau, i)$ denote the minimum cost of a path that starts from the depot and ends at vertex $i$, visiting each vertex in set $S$ exactly once, with an elapsed time $\tau$. $F(S, t, i)$ can be computed by solving the recurrence equation:
>
> \begin{equation}F(S, \tau, i) = \min_{(j,i)\in E}\\{F_i(S\setminus \\{i\\}, \tau^{\prime}, j) + c_{ji} | \tau+t_{ji}\leq w_i^2, w_j^1 \leq \tau^{\prime}\leq w_j^2\\}, \forall i\in V, S\subseteq V,   w_i^1 \leq \tau \leq w_i^2.  \quad \(1\)\end{equation}
>
> The set $S$ denotes the visited vertices, encoded as a binary vector of length $|V|$. $c_{ji}$ denotes the weight associated with the edge $e_{ji}$. The relaxation on elementarity restrictions reduces the state $(S, \tau, i)$ to a lower-dimensional state $(\Pi(P), \tau, i)$, where $\Pi(P)$ encoded as a binary vector of length $|N_i|\ll|V|$. The TSPTW shares the same state space $(S, \tau, i)$. However, their dominance rules are different: if state $(S, \tau, i)$ dominates another state $(S^\prime, \tau\prime, i)$, the TSPTW requires $S=S^{\prime}$, whereas the RCESPP is $S\subseteq S^{\prime}$. This is because RCESPP does not require visiting all vertices. Moreover, the edge weights in RCESPP are not restricted to distances and can even take negative values. Relaxing the elementarity restrictions in the TSPTW results only in self-loops at the depot or back-and-forth trips, which are neither feasible nor help tighten the relaxation. Therefore, our approach is not applicable to the TSPTW or the VRPTW.
>
> Nevertheless, based on a similar formulation, our algorithm can be readily extended to the generic Orienteering Problem [2] and the Capacitated Orienteering Problem (COP) [3]. However, the relaxation of elementarity constraints may be less effective for capacity constraints. When an unvisited vertex is not reachable due to time window constraint, the DP algorithm treats it as if it were already visited, which relaxes the dominance condition $S\subseteq S^{\prime}$ [4]. Moreover, tight time windows prevent the DP algorithm from generating too many loops along the path, thereby requiring small size of ng-sets.
>
> In conclusion, although the relaxation of elementarity restrictions is inherently problem-dependent, this work successfully demonstrates a novel paradigm that leverages learning-based prediction to alleviate the curse of dimensionality in dynamic programming. We believe that future researchers may develop alternative forms of state-space relaxation tailored to their specific problems and explore their integration with learning-based methods.
>
> **2. Test on difficult instances.**
>
> The SOTA pulse algorithm [5] is highly competitive for solving the OPTW or RCESPP, achieving more than a 70-fold speedup over the previous SOTA approach on Cordeau’s benchmark instances. Both the proposed DP-NG and its learning-enhanced variant, DP-NG-ML, exhibit clear performance improvements over the pulse algorithm. To evaluate performance on harder instances, we generated 20 Cordeau's instances with 100 vertices and time windows twice as wide. The computational times in seconds are shown below. However, DP-NG-ML does not performs well using the model trained on instances with regular time window instances. We are generating new 50-vertex instances with wide windows to retrain the model. As labeling is time-consuming, especially for hard instances, we will release the results of DP-NG-ML once they are ready.
>
> | Instance| OptVal | SOTA-pulse | DP-NG   \|| Instance| OptVal | SOTA-pulse | DP-NG    |
> |---------|--------|------------|---------|---------|--------|------------|----------|
> | 100_0   | 511    | 71.7  | 367.2    | 100_10  | 577    | 1022.0  | 1452.0  |
> | 100_1   | 485    | 19.2   | 161.3 | 100_11  | 538    | 10.3    | 92.1    |
> | 100_2   | 498    | 18.5  | 35.8    | 100_12  | 486    | 7.6     | 40.9    |
> | 100_3   | 482    | 13.6   | 13.5  | 100_13  | 482    | 46.5    | 66.8    |
> | 100_4   | 503    | 265.4  | 93.1   | 100_14  | 583    | 1031.1   | 97.2    |
> | 100_5   | 535    | 330.8  | 256.8  | 100_15  | 493    | 89.8    | 192.8   |
> | 100_6   | 516    | 8.1  | 11.7   | 100_16  | 462    | 6.9      | 10.9    |
> | 100_7   | 435    | 7.1   | 27.2   | 100_17  | 505    | 175.9    | 695.9   |
> | 100_8   | 495    | 24.2  | 148.3  | 100_18  | 488    | 2293.2   | 363.4   |
> | 100_9   | 460    | 4.7  | 13.4   | 100_19  | 463    | 31.5     | 38.3    |
> | Mean    |        |         |         |         |        | 273.9     | 208.9   |
>
> **3. Comparison with DiDPPy**
>
> We tested the DiDPPy solver on the OPTW using Cordeau instances. For a fair comparison, we added forced transitions that treat unvisited vertices as visited if they are infeasible due to time window constraints at the current state. In this case, more unpromising states will be pruned. Unfortunately, instances with more than 90 vertices ran out of memory. We select the first 90 vertices (excluding the depot) from each instance to create new instances. The computational time in seconds are shown below. The number of vertices is indicated in the name. The DP-NG is 100 times faster than the DiDPPy using the CABS solver. With the learning method integrated, the speedup reaches up to 800 times.
>
> | Instance | OptVal | DiDPPy  | DP-NG  | Speedup1 | DP-NG-ML | Speedup2 |
> |----------|--------|---------|--------|----------|----------|----------|
> | pr01_48  | 308    | 3.094   | 0.233  | 13.29    | 0.079    | 38.93    |
> | pr02_90  | 392    | 148.266 | 0.882  | 168.03   | 0.139    | 1065.18  |
> | pr03_90  | 331    | 51.841  | 0.782  | 66.29    | 0.231    | 224.82   |
> | pr04_90  | 356    | 33.328  | 0.674  | 49.43    | 0.169    | 197.45   |
> | pr05_90  | 443    | 193.404 | 1.386  | 139.54   | 0.551    | 350.70   |
> | pr06_90  | 329    | 29.954  | 0.527  | 56.85    | 0.046    | 656.43   |
> | pr07_72  | 298    | 12.956  | 0.187  | 69.14    | 0.016    | 810.99   |
> | pr08_90  | 374    | 19.59   | 0.397  | 49.30    | 0.034    | 568.74   |
> | pr09_90  | 320    | 100.259 | 0.346  | 289.93   | 0.057    | 1757.46  |
> | pr10_90  | 399    | 169.158 | 0.862  | 196.35   | 0.069    | 2465.47  |
> | Mean     |        | 76.185  | 0.628  | 109.81   | 0.139    | 813.62   |
>
> **4. The edge reduction procedure**
>
> The explanation of this procedure was placed in Appendix B.1. We will move it to the main text in the next revision.
> _"To improve prediction accuracy, we propose an edge reduction procedure that leverages problem-specific characteristics to exclude edges that cannot belong to ng-sets. Two types of edges are excluded: (1) the edges connected to the depot or self-loops; (2) any edge $e\_{ij}$ satisfying the following condition:_
> \begin{equation}
> \max \\{w_j^1 + d_j + t_{ji}, w_i^1\\} + d_i + t_{ij} > w_j^2.   \quad \(2\)
> \end{equation}
> _The first type of edges is excluded by definition, while the second type corresponds to a situation where cycle $(j-i-j)$ is infeasible due to the time window constraints."_
>
> This is a preprocessing procedure that does not involve learning. Since the other methods do not include ng-set prediction, this step is not required for them.
>
> **5. Sample complexity analysis**
>
> We appreciate the reviewer’s insightful comment. While we have not performed a formal sample complexity analysis, our empirical experience suggests that around 10,000 training instances typically yield satisfactory predictive accuracy, with more data further improving performance. We will conduct additional experiments and include the results in the appendix of the revised version.
>
> **Reference**
>
> [1] G. Righini and M. Salani. Decremental state space relaxation strategies and initialization heuristics for solving the orienteering problem with time windows with dynamic programming. Computers & Operations Research, 36(4):1191–1203, 2009.
>
> [2] M. Fischetti, J. J. S. Gonz´alez, and P. Toth. Solving the orienteering problem through branch-and-cut. INFORMS Journal on Computing, 10(2):133–148, 1998.
>
> [3] A. Bock and L. Sanit`a. The capacitated orienteering problem. Discrete Applied Mathematics, 195:31–42, 2015.
>
> [4] D. Feillet, P. Dejax, M. Gendreau, and C. Gueguen. An exact algorithm for the elementary shortest path problem with resource constraints: Application to some vehicle routing problems. Networks, 44(3):216–229, Oct. 2004.
>
> [5] D. Duque, L. Lozano, and A. L. Medaglia. Solving the Orienteering Problem with Time Windows via the Pulse Framework. Computers & Operations Research, 54:168–176, 2015.

---

> > ### Author Response · Authors · 2025-08-05
> > **Complete results on difficult instances with wider time windows.**
> >
> > | Instance | OptVal | SOTA-pulse | DP-NG ngSize | DP-NG iter | DP-NG time | DP-NG-ML ngSize | DP-NG-ML iter | DP-NG-ML time | Speedups |
> > |----------|--------|------------|--------------|------------|------------|-----------------|---------------|---------------|----------|
> > | 100_0    | 511    | 71.77      | 5.56         | 67         | 367.28     | 6.99            | 12            | 281.19        | 1.31     |
> > | 100_1    | 485    | 19.29      | 5.79         | 67         | 161.33     | 6.51            | 18            | 113.25        | 1.42     |
> > | 100_2    | 498    | 18.51      | 3.55         | 38         | 35.85      | 5.92            | 2             | 26.15         | 1.37     |
> > | 100_3    | 482    | 13.66      | 2.76         | 33         | 13.53      | 4.98            | 4             | 8.39          | 1.61     |
> > | 100_4    | 503    | 265.44     | 4.84         | 53         | 93.12      | 6.85            | 12            | 112.46        | 0.83     |
> > | 100_5    | 535    | 330.85     | 5.22         | 49         | 256.83     | 6.49            | 2             | 137.34        | 1.87     |
> > | 100_6    | 516    | 8.17       | 2.7          | 25         | 11.78      | 5.6             | 1             | 6.86          | 1.72     |
> > | 100_7    | 435    | 7.18       | 3.72         | 45         | 27.26      | 5.05            | 8             | 32.86         | 0.83     |
> > | 100_8    | 495    | 24.29      | 4.91         | 58         | 148.31     | 6.25            | 3             | 33.03         | 4.49     |
> > | 100_9    | 460    | 4.75       | 2.95         | 35         | 13.47      | 4.52            | 6             | 4.85          | 2.78     |
> > | 100_10   | 577    | 1022.04    | 7.35         | 83         | 1452.04    | 8.49            | 16            | 1276.55       | 1.14     |
> > | 100_11   | 538    | 10.31      | 4.14         | 54         | 92.12      | 6.69            | 6             | 86.07         | 1.07     |
> > | 100_12   | 486    | 7.60       | 5.02         | 51         | 40.95      | 6.63            | 11            | 45.22         | 0.91     |
> > | 100_13   | 482    | 46.58      | 4.12         | 49         | 66.83      | 5.2             | 7             | 38.31         | 1.74     |
> > | 100_14   | 583    | 1031.12    | 4.14         | 39         | 97.24      | 7.45            | 2             | 129.05        | 0.75     |
> > | 100_15   | 493    | 89.88      | 4.2          | 60         | 192.83     | 6.21            | 5             | 235.07        | 0.82     |
> > | 100_16   | 462    | 6.90       | 3.3          | 34         | 10.92      | 4.79            | 7             | 6.14          | 1.78     |
> > | 100_17   | 505    | 175.96     | 5.81         | 66         | 695.98     | 6.41            | 16            | 450.45        | 1.55     |
> > | 100_18   | 488    | 2293.26    | 6.54         | 56         | 363.42     | 6.82            | 6             | 149.93        | 2.42     |
> > | 100_19   | 463    | 31.56      | 4.15         | 56         | 38.37      | 5.01            | 16            | 20.67         | 1.86     |
> > | Mean     |        | 273.96     | 4.54         | 50.9       | 208.97     | 6.14            | 8.0           | 159.69        | 1.61     |
> >
> >
> > The results show that the learning-based method accelerates the DP-NG algorithm by a factor of 1.61 on average, while still significantly outperforming the state-of-the-art pulse algorithm. The improvement is limited by the accuracy of prediction, which can be further improved by training the model with more training instances or using larger instances with 100 vertices.

---

### Official Review · Reviewer_Nr8A · 2025-07-07

**Clarity:** 2
**Significance:** 1
**Originality:** 1
**Rating:** 2
**Confidence:** 3

**Summary:**

This paper considers Orienteering Problem with Time Windows (OPTW), a more practical and harder variant of the traveling salesman problem. The paper considers the "ng-route relaxation method", which is a dynamic programming algorithm that solves the problem exactly. The idea is that optimal solutions are found, that maybe contain cycles. Gradually, the restrictions are increased to ensure that a feasible solution is eventually found. The original method is to simply add existing cycles of the current optimal solution to the "ng-sets" of each vertex to eliminate these in the next round. The authors propose a deep-learning method, based on Graph Convolutional Networks (GCN) to try to speed up this algorithm by predicting which edgess should belong to the ng-sets. A modification of the original GCN model to directed graphs is proposed.

**Questions:**

The context of the literature provided in the paper is very minimal. If the authors could greatly expand this discussion, so that the novelty of the proposed method could be accurately assessed, it could impact my rating. See more discussion under Weaknesses.

**Ethical Concerns:**

["NO or VERY MINOR ethics concerns only"]

**Final Justification:**

I have read the authors' rebuttal and the other reviews.
In response:
- A comprehensive literature survey of ML for combinatorial optimization problems is not required. However, it remains important to place the proposed methods in context and clearly explain the novelty. For example, in the contributions paragraph, there is this sentence (line 71):  "This innovative paradigm is anticipated to generalize across a broad spectrum of COPs defined over graph structures, such as the Vehicle Routing Problem (VRP) [ 2], the delivery man problem [ 25 ], and others." But the related work section discusses the OPTW problem only. Relevant work on related problems should be discussed.

- The way the paper was written, I was under the impression the Section 4.2 was a significant component of the novelty of the proposed method (Indeed, it appears to be the only technological innovation introduced). The main argument for novelty is the generalization of DiConvNet to directed graphs. My point was that there are many directed GNN architectures that already exist that may perform well. In the rebuttal, the authors indicate that Section 4.2 is not a major part of their contribution.

- Finally, after reading the other reviewer comments, I agree that problem instances that can be solved exactly in seconds are probably too small to demonstrate the effectiveness of the proposed method.

**Limitations:**

yes

**Quality:**

2

**Strengths And Weaknesses:**

+ Speeding up exact algorithms using ML is a good role for ML within combinatorial optimization, as the authors argue, as the optimality guarantee of the original, exact algorithm still holds. However, this is far from the first work to propose this idea.
+ The proposed method appears to work well in practice, as evidenced by the empirical results.

Weaknesses:
- There are quite a few works on using deep learning, including RL and GNNs, to accelerate the solution of linear programs. Given the connection between dynamic programming and LPs, I wonder how relevant this literature could be to this paper. For example, here's a survey for ML-augmented approaches to branch-and-bound [1]. Very little related work, concerning ML to speed up exact optimization, is discussed by the authors.

[1] Scavuzzo, L., Aardal, K., Lodi, A. et al. Machine learning augmented branch and bound for mixed integer linear programming. Math. Program. (2024). https://doi.org/10.1007/s10107-024-02130-y

- There are many works adapting the original GCNs to directed graphs. So it's unclear what the novelty here is. I'm hardly an expert in this area, but the architecture proposed in this paper doesn't seem very novel. Moreover, none of this related work is discussed, and it is presented a major contribution to adapt the GCN to directed graphs.

For example:

[2] Korban and Li. DDGCN: A Dynamic Directed Graph Convolutional Network for Action Recognition.
[3] Rossi et al. Edge Directionality Improves Learning on Heterophilic Graphs.

---

> ### Author Rebuttal · Authors · 2025-07-30
>
> Thank you for your valuable comments and suggestions. Owing to the strict page limit, we did not include a comprehensive review of the literature on solving combinatorial optimization problems (COPs) with machine learning (ML). We will expand this part in the revised version.
>
> **1. Positioning of this work in related literature.**
>
> The use of ML to solve COPs has emerged as a highly promising research direction, attracting considerable attention in recent years. Most existing approaches employ ML models as heuristics to directly solve the problems by constructing satisfactory solutions via learned predictions. However, these methods are often detached from the underlying mathematical models, lacking theoretical guarantees on solution quality.
>
> Research on integrating ML with exact methods for COPs remains relatively limited, largely due to the complexity of designing such algorithms and the black-box nature of learning models. The predominant research trend is to formulate COPs as mixed-integer programming models (MIP models) and enhance key components of traditional exact approaches for MIP models using ML techniques. The majority of research efforts concentrate on enhancing branch-and-bound (B\&B) based algorithms, which serve as the foundational theoretical tool in modern MILP solvers [1]. In this line of research, ML techniques are leveraged to guide primal heuristics, make high-quality branching decisions, separate cutting planes to strengthen the formulation, and improve node selection strategies within B\&B frameworks. These enhancements are problem-agnostic and can be directly integrated into generic MIP solvers.
>
> Our work focuses on a deep integration of ML techniques with dynamic programming (DP), another exact solving paradigm for COPs. Unlike MIP-based approaches, DP solves COPs by explicitly defining a state-space formulation and determining state transition recurrences based on the problem’s specific structure. DP is widely used to efficiently solve COPs with clear problem structures, such as knapsack and routing problems. However, its main drawback is the risk of state space explosion, also known as the `curse of dimensionality'. To address this weakness, Christofides et al. [2] presented a state space relaxation technique which projects the search space explored by DP onto a lower-dimensional space to accelerate the DP. The main drawback of such relaxation is that the search in the lower dimensional state space does not guarantee to find an optimal solution. A practical compromise is to iteratively tighten the relaxation based on the computed infeasible solution, until an optimal solution is eventually found [3, 4] However, this iterative framework may lead to higher computational costs, as the DP algorithm needs to be executed multiple times, and the final relaxation is only identified once the computed solution is feasible.
>
> To overcome this limitation, we utilize ML techniques to predict the final relaxation, so that the DP algorithm can achieve optimality in low-dimensional state space with limited iterations and computation time. In essence, the learning model approximates a projection function that maps the original state space to an appropriate lower-dimension space. To the best of our knowledge, this learning-based paradigm for COPs has not been explored in the existing literature.
>
> This work takes the OPTW as an example to demonstrate such paradigm. The DP algorithm for solving this problem operates in a high-dimensional state space $(S, \tau, i)$, where $S$ represents the set of visited vertices, $\tau$ denotes the elapsed time, and $i$ is the associated vertex. The size of the state-space graph is heavily influenced by $S$ as it is typically encoded as a binary vector of length $|V|$. Therefore, we employ the ng-route relaxation technique [5] to relax the state space onto a lower-dimensional space $(\Pi(P), \tau, i)$ where $\Pi(P)$ can be encoded as a binary vector of length $|N_i| (|N_i|\leq |V|)$. The detailed procedure is explained in Section 3 and Appendix A. The proposed learning model approximates the ng-set $N_i$ for each vertex $i$, enabling the DP algorithm to operate in a customized state space and efficiently find the optimal solution.
>
> **2. The learning model.**
>
> This work does not aim to introduce a sophisticated GCN model designed for general-purpose learning on directed graphs. Instead, we adopt a learning model that is sufficient for the prediction tasks required by our framework. Specifically, we select the Residual Gated GCN, a simple, scalable, and well-studied architecture. It has demonstrated competitive performance on the Traveling Salesman Problem (TSP) [6], which shares structural similarities with the OPTW. We adapt this model to a directed version to suit our prediction tasks, and the resulting performance remains competitive when compared to the state-of-the-art exact algorithm.
>
> In preliminary experiments, we also evaluated more advanced architectures, such as Transformer-based models implemented with the PyTorch Geometric library. However, they exhibited poor performance in our setting, likely due to the need for further adaptations.  While we acknowledge that there may exist more suitable and well-developed learning models for our task, a comprehensive exploration of such models is beyond the scope of this work.
>
> **3. Conclusion.**
>
> In summary, we present a novel paradigm that integrates DP with modern ML techniques for solving COPs. The numerical results demonstrate the effectiveness and superiority of the proposed learning-based enhancements. This learning approach can be generalized to other COPs by using DP with problem-specific state space relaxation. We believe that this work will serve as a valuable starting point to advance the deep integration between operations research and artificial intelligence.
>
> **Reference**
>
> [1] L. Scavuzzo, K. Aardal, A. Lodi, and N. Yorke-Smith. Machine learning augmented branch and bound for mixed integer linear programming. Mathematical Programming, 2024.
>
> [2] N. Christofides, A. Mingozzi, and P. Toth. State-space relaxation procedures for the computation of bounds to routing problems. Networks, 11(2):145–164, 1981.
>
> [3] N. Boland, J. Dethridge, and I. Dumitrescu. Accelerated label setting algorithms for the elementary resource constrained shortest path problem. Operations Research Letters, 34(1):58–68, 2006.
>
> [4] G. Righini and M. Salani. Decremental state space relaxation strategies and initialization heuristics for solving the orienteering problem with time windows with dynamic programming. Computers & Operations Research, 36(4):1191–1203, 2009.
>
> [5] R. Baldacci, A. Mingozzi, and R. Roberti. New Route Relaxation and Pricing Strategies for the Vehicle Routing Problem. Operations Research, 59(5):1269–1283, 2011.
>
> [6]  C. K. Joshi, T. Laurent, and X. Bresson. An efficient graph convolutional network technique for the travelling salesman problem, 2019.

---

> ### Comment · Reviewer_Nr8A · 2025-08-06
>
> Thank you for the response. I have read the authors' rebuttal and the other reviews.
> In response:
> - A comprehensive literature survey of ML for combinatorial optimization problems is not required. However, it remains important to place the proposed methods in context and clearly explain the novelty. For example, in the contributions paragraph, there is this sentence (line 71):  "This innovative paradigm is anticipated to generalize across a broad spectrum of COPs defined over graph structures, such as the Vehicle Routing Problem (VRP) [ 2], the delivery man problem [ 25 ], and others." But the related work section discusses the OPTW problem only. Relevant work on related problems should also be discussed.
>
> - The way the paper was written, I was under the impression the Section 4.2 was a significant component of the novelty of the proposed method (Indeed, it appears to be the only technological innovation introduced). The main argument for novelty is the generalization of DiConvNet to directed graphs. My point was that there are many directed GNN architectures that already exist that may perform well. In the rebuttal, the authors indicate that Section 4.2 is not a major part of their contribution.
>
> - Finally, after reading the other reviewer comments, I agree that problem instances that can be solved exactly in seconds are probably too small to demonstrate the effectiveness of the proposed method.

---

> > ### Author Response · Authors · 2025-08-07
> > **Response to the contribution**
> >
> > Thank you for the valuable comments. We would like to repsonse them one by one:
> > 1. As the title suggests, this work focuses on a specific NP-hard problem: the Orienteering Problem with Time Windows (OPTW). We do not aim to generalize the proposed method to other problems in this paper. As discussed in our response to Reviewer Ax3s, the Vehicle Routing Problem (VRP) and OPTW differ significantly. However, the state-of-the-art branch-and-price algorithms for VRP often use the OPTW (or RCESPP) as a subproblem and apply ng-route relaxation as a core technique. These connections indicate potential for extending our methodology to VRP and other related problems, with suitable problem-specific adaptations. We are not certain whether a detailed explanation of the connection between the OPTW and other related problems is appropriate for the literature review section, as not all readers may be experts in routing problems. Instead, we hope that this work can inspire other researchers to adopt and adapt this paradigm to their own specific problems.
> > 2. We did not emphasize in the paper that DiConvNet is our main contribution. As stated in the methodological innobation: "Our method combines the strengths of machine learning and operations research, preserving optimality while significantly enhancing performance through data-driven insights...... To the best of our knowledge, this is the first work to propose such a solution paradigm. " Existing directed GCNs have not demonstrated the same effectiveness in solving combinatorial optimization problems as the Residual Gated GCN. Furthermore, the provided directed GCNs do not incorporate edge features or exhibit strong scalability across different problems. As clarified earlier, identifying the best architecture is not the core novelty of this work.
> > 3.  As noted in our rebuttal to Reviewer Ax3s, the problem instances can be solved to optimality within a few seconds because the SOTA pulse algorithm we compared against is extremely competitive. Outperforming such algorithm highlights the efficiency of our learning-augmented method. Moreover, we have provided comparative results on harder instances with wider time windows, which require over 200 seconds to solve.

---

### Note · Authors · 2025-08-13

This work presents a novel paradigm that leverages modern machine learning to address combinatorial optimization problems. While most prior AI-related research has focused on heuristics or enhancing exact MIP techniques, our approach aims at accelerating dynamic programming by learning the problem-specific state-space relaxations, thereby enabling dimensionality reduction in the dynamic programming process. To the best of our knowledge, this form of integrating AI with OR has not been previously explored.

While the learning model applied might not be the most complicated, our method has demonstrated superior performance compared with the state-of-the-art. This significant improvement stems from the deep neural network's ability to learn problem-specific constraints, unlike prior work that prioritized generality and model-agnostic characteristics. To generalize this idea, future researchers may develop learning-augmented state-space relaxation methods tailored to their specific combinatorial optimization problems.

---

### Decision · Program_Chairs · 2025-09-17

**Decision:**

Accept (poster)

**Comment:**

The paper augments a dynamic programming formulation for the Orienteering Problem with Time Windows by edge selection for a combinatorial structure (ng-sets) that is used in the DP base algorithm. Reviewers acknowledge that the paper is well written and that the improvements are significant. On the other hand, methodological contribution is criticized as well as practical relevance, since the method was evaluated on a very specific problem formulation only, hence generalization to other problems is open, and additionally all instances can be solved quickly with established classical algorithms. Also, as another disadvantage, the method is supervised, hence ground truth labels must be computed first.
The rebuttal has addressed partially the issue of generalizability to other problems. On the topic of relevance, it is positively noted that the machine learning component is embedded in an exact algorithm, hence not sacrificing solution quality for faster runtime and making it an overall elegant approach. As such, even though reviewers have mixed feelings and there remained some disagreement, the paper merits acceptance at NeurIPS.